**Resource**

# Mime-seq 2.0: a method to sequence microRNAs from specific mouse cell types

Ariane Mandlbauer [1,2], Qiong Sun [2], Niko Popitsch [3,4], Tanja Schwickert [2], Miroslava Spanova [2], Jingkui Wang[2], Stefan L Ameres [3,4 ✉], Meinrad Busslinger [2 ✉] & Luisa Cochella [1,2 ✉]

## Abstract

**Many microRNAs (miRNAs) are expressed with high spatio-temporal specificity during organismal development, with some being limited to rare cell types, often embedded in complex tissues. Yet, most miRNA profiling efforts remain at the tissue and organ levels. To overcome challenges in accessing the microRNomes from tissue-embedded cells, we had previously developed mime-seq (miRNome by methylation-dependent sequencing), a technique in which cell-specific miRNA methylation in *C. elegans* and *Drosophila* enabled chemo-selective sequencing without the need for cell sorting or biochemical purification. Here, we present mime-seq 2.0 for profiling miRNAs from specific mouse cell types. We engineered a chimeric RNA methyltransferase that is tethered to Argonaute protein and efficiently methylates miRNAs at their 3′-terminal 2′-OH in mouse and human cell lines. We also generated a transgenic mouse for conditional expression of this methyltransferase, which can be used to direct methylation of miRNAs in a cell type of choice. We validated the use of this mouse model by profiling miRNAs from B cells and bone marrow plasma cells.**

**Keywords** miRNA Methylation; miRNA Profiling; Rare Cell Types; Mammalian System; B Cells
**Subject Categories** Methods & Resources; RNA Biology

## Introduction

MicroRNAs (miRNAs) are short (21–23 nt) non-coding RNAs that act as post-transcriptional repressors and play important roles in animal physiology and development (Bartel, 2018). The expression patterns of individual miRNAs have been most systematically studied by using in situ hybridization or transcriptional reporters in model organisms, revealing that many miRNAs are expressed with high cell-type specificity (Wienholds et al, 2005; Aboobaker et al, 2005; Martinez et al, 2008; Landgraf et al, 2007; Park et al, 2012;

Alberti et al, 2018; Alberti and Cochella, 2017). Despite this, expression profiling using sequencing to uncover the microRNome in a sample of interest is typically done on whole organisms, organs or sections of tissue, missing the cellular resolution required to fully understand the function of any given miRNA. The importance of addressing such cellular heterogeneity has been the driving force to establish single-cell messenger RNA (mRNA) sequencing approaches. However, because of challenges associated with the short nature of miRNAs, single-cell profiling methods have not yet been widely adapted for small RNAs. Instead, profiling of miRNAs from individual cell types has mainly relied on two different approaches. First, the sorting or microdissection of the cells of interest for subsequent RNA extraction and small RNA library preparation (e.g., Jenike et al, 2023; Grolmusz et al, 2016). Second, the cell-specific expression of an epitope-tagged Argonaute protein, into which miRNAs are loaded, that can be immunoprecipitated to retrieve the associated small RNAs (Than et al, 2013; He et al, 2012). All those approaches have been successfully used in a number of scenarios. However, they also present some technical challenges and may not be straightforward to implement in all cases. For laser microdissections and FACS-based cell sorting, cells of interest are physically isolated prior to preparation of small RNA libraries. Long procedures that include tissue fixation or cell dissociation are required for sample preparation. Both isolation approaches are technically challenging and require complex equipment and technical expertise. When working with rare cell types or lowly abundant miRNAs, low yields as well as contaminants from surrounding cells might decrease the signal-to-noise ratio drastically and produce false-positive results. On the other hand, protocols for immunopurification of epitope-tagged AGO require relatively high input for good quality data and rely on overexpression of Argonaute, which might not be desirable in certain cell types (Reichholf et al, 2019).

To overcome some of these challenges and to provide an alternative approach, we previously developed mi*RNome by* me*thylation dependent sequencing* (mime-seq) to retrieve the miRNomes of specific-cell types within developing *C. elegans* or *D. melanogaster* (Alberti et al, 2018). Mime-seq relies on the fact that animal miRNAs are not normally methylated at their 3′ end and thus transgenic expression of a methyltransferase in a cell-specific manner

[1]School of Medicine, John Hopkins University, Baltimore, MD, USA. [2]Research Institute of Molecular Pathology (IMP), Vienna BioCenter (VBC), Vienna, Austria. [3]Max Perutz Labs (MPL), Vienna BioCenter (VBC), Vienna, Austria. [4]University of Vienna, Center for Molecular Biology, Department of Biochemistry and Cell Biology, Vienna, Austria. ✉E-mail: stefan.ameres@univie.ac.at; meinrad.busslinger@imp.ac.at; mcochel1@jhmi.edu

deposits a distinguishing mark on miRNAs derived from the cell type of interest. Methylated miRNAs are then selectively detected by performing an oxidation treatment prior to standard small RNA cloning and sequencing. Oxidation by NaIO$_4$ renders the terminal ribose of unmethylated miRNAs non-ligatable while the presence of 2′OMe prevents oxidation and leaves the 3′OH intact and available for adapter ligation and subsequent sequencing (Fig. 1A). We first established mime-seq by taking advantage of the fact that plant miRNAs are endogenously methylated by HEN1 and specifically, we

used HEN1 from *Arabidopsis thaliana* (At-HEN1) (Yu et al, 2005; Yang et al, 2006) to ectopically methylate miRNAs in animals (Alberti et al, 2018). With its two dsRNA-binding domains, At-HEN1 recognizes the products of Dicer cleavage and positions the 3′ 2-nt overhangs of these 21–23 nt long RNA duplexes in the catalytic domain for 2′O-methylation (Huang et al, 2009). Transgenic expression of At-HEN1 under cell-specific promoters in worms and flies resulted in efficient cell-specific miRNA methylation. Because mime-seq proved to be useful for miRNA profiling in

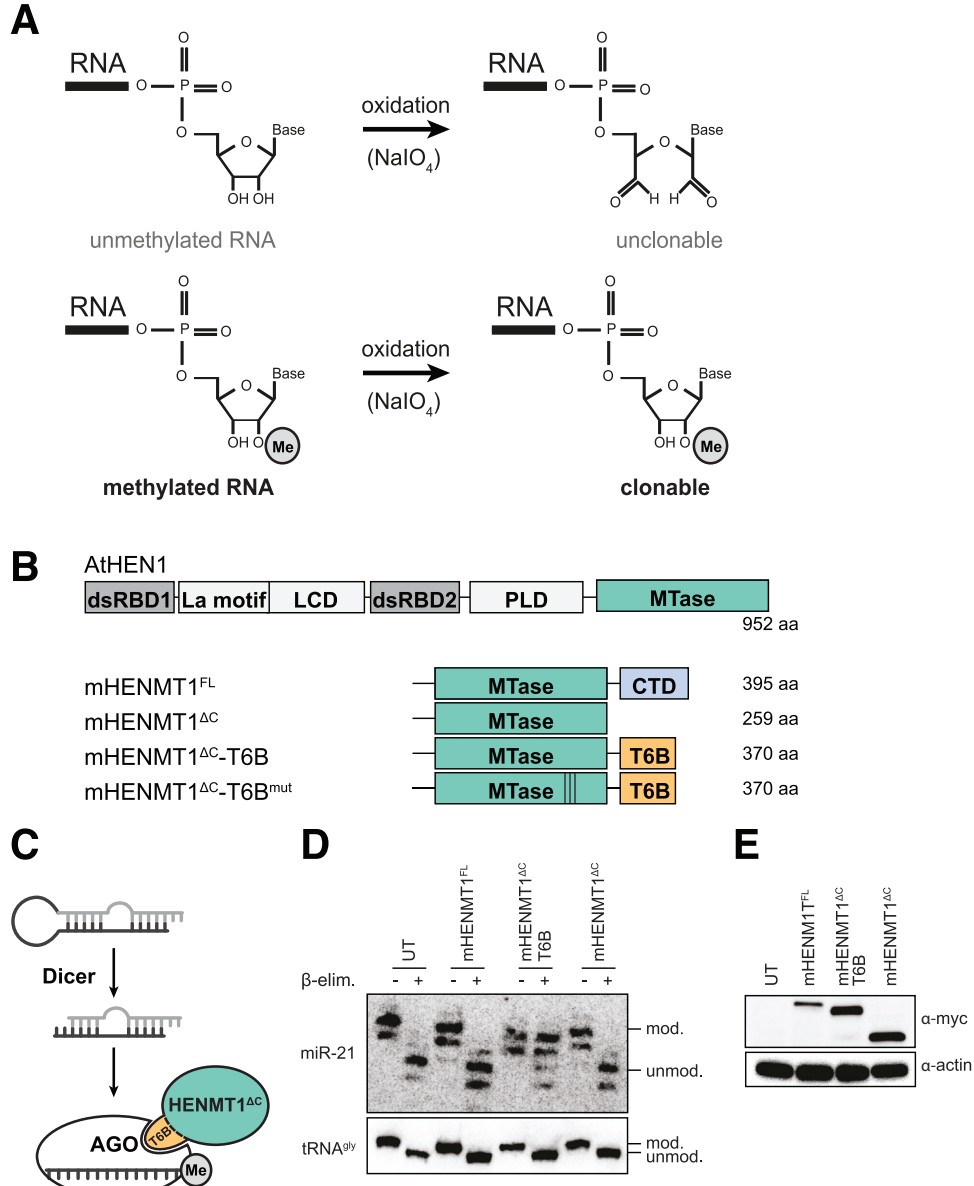

**Figure 1. HENMT1$^{\Delta C}$-T6B, an engineered methyltransferase, efficiently methylates mouse miRNAs in cultured cells.**

(A) Schematic of the reaction enabling selective adapter ligation of methylated RNA. Unmethylated small RNAs are oxidized by NaIO$_4$, 2′O-methylated miRNAs are protected and retain a free 3′OH, required for 3′ adapter ligation. (B) Overview of RNA methyltransferases designed and used in this study. (C) Schematic diagram of T6B-dependent tethering of HENMT1$^{\Delta C}$ to AGO. (D) Murine embryonic fibroblasts (MEFs) carrying lentivirus-mediated integration of the indicated transgenes under a SFFV promoter. Total RNA extracted was subjected to oxidation and β-elimination and the products resolved by high-resolution northern blotting with the indicated probes (tRNA$^{Gly}$ was used to monitor oxidation completion, and as loading control). "+" refers to oxidized and β-eliminated samples, "−" to untreated controls. Position of methylated (mod.) and unmethylated (unmod.) RNAs is indicated. UT: untransduced cells. (E) Corresponding western blot for experiment in (D) to detect the various Myc-tagged enzyme variants. Actin served as loading control. Experiments were performed at least twice per cell line. Source data are available online for this figure.

*C. elegans* and *Drosophila*, we set out to adapt it for use in mammalian systems, specifically in mice.

Here, we report mime-seq 2.0 for implementation in mice, featuring an engineered methyltransferase and a transgenic mouse for its conditional, cell-specific expression. Surprisingly, in our initial attempts to perform mime-seq in mammalian cells, we found that At-HEN1 does not efficiently methylate miRNAs in mouse or human cells. This led us to develop an alternative methyltransferase. We reasoned that tethering an RNA methyltransferase domain to the Argonaute proteins in which miRNAs are loaded would bring the methylation activity to the proximity of the desired substrates. We thus started with the piRNA 2′O-methyltransferase from mouse, HENMT1, and modified it by removing its C-terminal domain that mediates interaction with the Piwi protein, and by adding instead a peptide derived from the Argonaute interactor TNRC6B/GW182. An 84-amino acid long peptide derived from this protein (T6B peptide) has been shown to directly interact with all four mouse Argonaute proteins (Hauptmann et al, 2015). We first show that the resulting chimeric enzyme, HENMT1$^{\Delta C}$-T6B, efficiently methylates miRNAs in cultured mouse and human cell lines. We report the generation and implementation of a transgenic mouse for conditional expression of this enzyme and show that it methylates miRNAs efficiently in B cells and plasma cells, without affecting B cell maturation. Lastly, mime-seq 2.0 is able to retrieve cell-specific miRNAs from cells present in a mix in a ratio of as little as 1 in 1000 cells. Taken together, mime-seq 2.0 provides a simplified approach that overcomes some of the current challenges towards achieving systematic cell-type-specific profiling of miRNAs in vivo in the mouse.

# Results

## At-HEN1 does not methylate miRNAs in cultured mammalian cells

To assess the methylation potential of At-HEN1 in mammalian cells, we transfected codon-optimized and Myc-tagged At-HEN1 into human HEK293T cells and assessed miRNA methylation by oxidation of the terminal ribose followed by β-elimination. This treatment removes the 3′-terminal nucleoside of unmethylated but not of methylated miRNAs and causes accelerated migration in high-resolution urea-PAGE that can be visualized by northern blotting (Alefelder, 1998). In contrast to what we had seen in worms and flies, we were unable to detect 2′O-methylation of endogenous miRNAs in HEK293T cells, despite detectable expression of At-HEN1 (Fig. EV1A,B). This was not because the enzyme made in these cells was inactive, as immunopurified Myc-At-HEN1 from HEK293T was fully active in an in vitro methylation assay in which we provided a radiolabeled duplex substrate (Fig. EV1C). This suggested that the difference in At-HEN1 activity in worms and flies versus human cells may reflect different availability of the At-HEN1 substrate, the miRNA duplex generated by Dicer. While somewhat surprising, this is consistent with the fact that in mammals, but not in worms or flies, Dicer cleavage is coupled to Argonaute loading (Maniataki and Mourelatos, 2005; Wang et al, 2009) and thus the RNA duplex may not be available for binding At-HEN1. To circumvent this, we engineered an alternative methyltransferase to meet the mechanistic requirements of the miRNA biogenesis pathway in mammals.

## The engineered HENMT1$^{\Delta C}$-T6B methyltransferase methylates miRNAs in cultured mouse and human cells

To develop a suitable methyltransferase, we decided to modify *Mus musculus* HENMT1. mHENMT1 is a methyltransferase specifically expressed in mouse testis (Kirino and Mourelatos, 2007b), where it is required for piRNA 2′O-methylation at the 3′ terminal ribose (Kirino and Mourelatos, 2007a; Ohara et al, 2007). Specificity for piRNA methylation is determined by the C-terminal domain of mHENMT that directly interacts with Piwi proteins in which piRNAs are loaded (Saito et al, 2007). Contrary to At-HEN1, substrates of mHENMT1 are single-stranded RNAs of varying lengths (Peng et al, 2018). To repurpose mHENMT1 for miRNA methylation, we removed the C-terminal domain and replaced it with a peptide that binds to the Argonaute (AGO) proteins that load miRNAs (Fig. 1B). The GW/TNRC6 family of proteins bind AGO to mediate its repressive function (Baillat and Shiekhattar, 2009). An 84-amino acid peptide derived from TNRC6B, referred to as T6B peptide, has been shown to effectively interact with all four mouse AGOs (Hauptmann et al, 2015). We hypothesized that tethering HENMT1 to AGO via the T6B peptide might force methylation of loaded miRNAs (Fig. 1C). We tested the activity of full-length mHENMT1 (HENMT1$^{FL}$), a C-terminal truncation mutant (HENMT1$^{\Delta C}$) and the same mutant fused to T6B (HENMT1$^{\Delta C}$-T6B) by stable integration in mouse embryonic fibroblasts (MEF) and transient transfection in HEK293T cells. Using β-elimination and high-resolution northern blots, we observed that only HENMT1$^{\Delta C}$-T6B, but not a catalytic mutant or any of the non-T6B versions, methylated miRNAs in MEFs (Fig. 1D,E) and in HEK293T cells (Fig. EV1D,E). During these experiments, we noticed variable expression levels of the enzymes, with HENMT1$^{\Delta C}$ and HENMT1$^{\Delta C}$-T6B having higher expression level than the others (Figs. 1E and EV1D). To exclude that expression levels alone determine the large differences in activity, we cloned all HENMT1s under a doxycycline-inducible TRE3 promoter and integrated them using a lentiviral system into a version of the human cell line RKO that expresses the rTA3 transactivator. After titrating doxycycline levels to achieve similar expression levels, still only HENMT1$^{\Delta C}$-T6B showed detectable methylation activity (Fig. EV1F,G). We note that codon-optimized At-HEN1 could never be expressed at levels close to mHENMT1, and we can thus not exclude that if high enough levels of this enzyme were achieved, it might show in vivo activity in mammalian cells. In sum, we designed a new chimeric enzyme that efficiently methylates miRNAs in human (HEK293T and RKO) and mouse (MEF) cells in culture.

## HENMT1$^{\Delta C}$-T6B enables unbiased retrieval of the miRNome by sequencing

To characterize the methylation activity of HENMT1$^{\Delta C}$-T6B across the microRNome, we took advantage of the doxycycline-inducible system in RKO cells and performed sequencing of oxidized and unoxidized small RNAs at different times post expression induction (Figs. 2A,B; Dataset EV1). We used a series of methylated small RNA spike-ins in known amounts, spanning a range of almost four orders of magnitude, for normalization (Lutzmayer et al, 2017) (see Methods). Spike-ins were added based on total RNA quantification before each sample was split in two, and one half was oxidized and

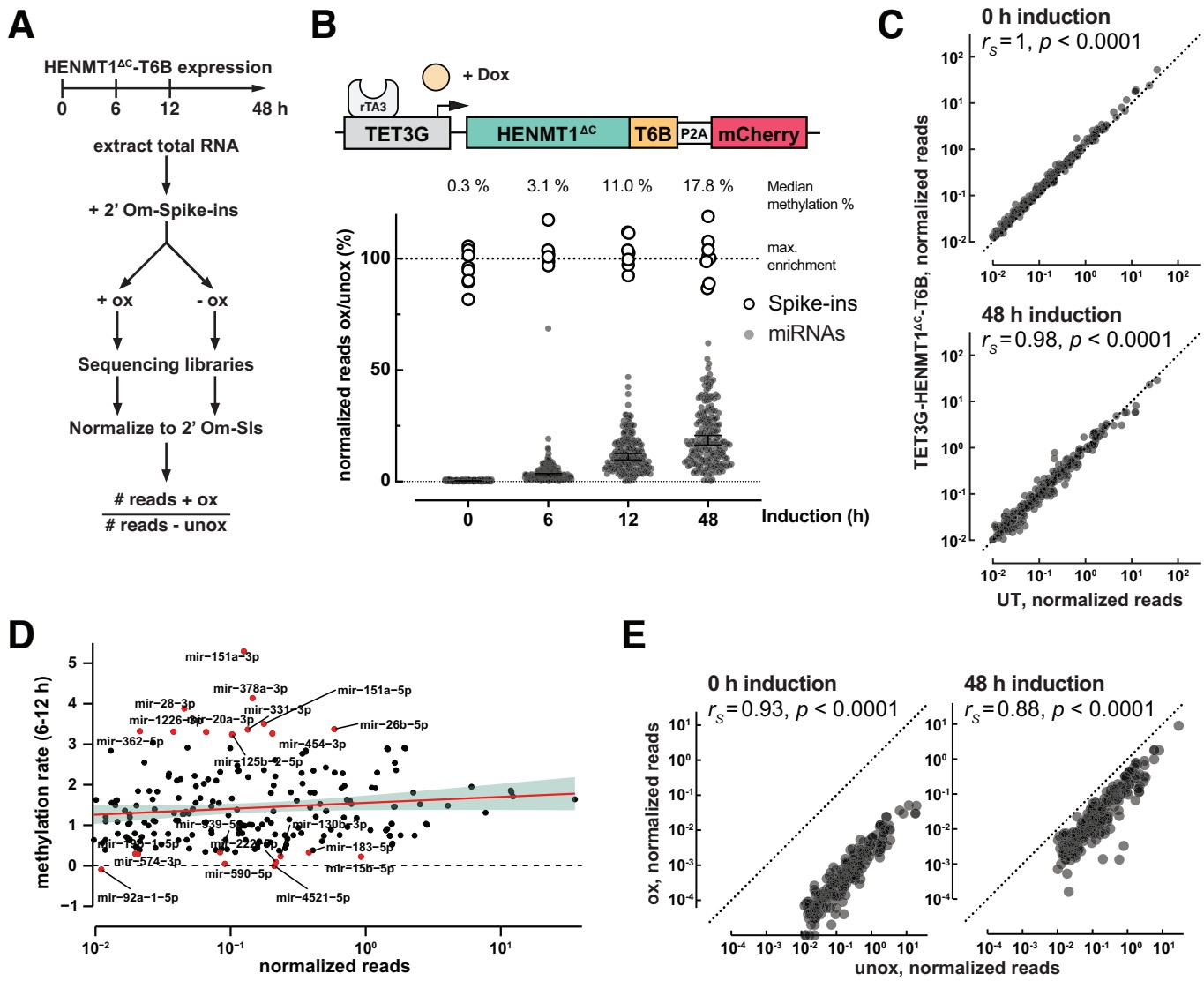

**Figure 2. Timecourse analysis of 2′O-methylation using inducible HENMT1ᐃC-T6B expression in RKO cells.**

(A) Schematic overview of the timecourse experiment. Total RNA was extracted from cells after 0–48 h of HENMT1ᐃC-T6B expression. After addition of methylated spike-ins samples are treated with or without $NaIO_4$ followed by generation of sequencing libraries. To investigate what fraction of miRNAs are methylated, we calculated the ratio of normalized reads between oxidized and unoxidized samples. (B–E) RKO cells with lentivirus-mediated integration of the rtTA3 trans-activator and inducible HENMT1ᐃC-T6B were used for a 48 h doxycycline induction time course. Total RNA was extracted at indicated timepoints, methylated spike-ins added, and oxidized and unoxidized samples subjected to small RNA sequencing. MicroRNAs with spike-in-normalized reads <0.01 were removed from the analysis. Sequencing libraries were prepared from one biological replicate. (B) Time-dependent increase of %ox/unox for miRNAs. Fully methylated spike-ins define the maximum possible enrichment. Lines indicate median enrichment with 95% CI. $N = 217$ miRNAs per timepoint. (C) Comparison of miRNA levels in unoxidized samples with or without HENMT1ᐃC-T6B. Expression of HENMT1ᐃC-T6B does not affect miRNA abundance ($r_S$ = Spearman correlation coefficient, UT: untransduced cells). (D) Methylation rate, calculated as the slope of % ox/unox between 6 and 12 h of induction, does not correlate with miRNA abundance. miRNAs with the top and bottom 5th percentile methylation rates are colored. (E) Normalized reads from oxidized vs. unoxidized samples at 0 and 48 h timepoints. The relative abundance of miRNAs is maintained after oxidation ($r_S$ = Spearman correlation coefficient).

the other half was not (Fig. 2A). The ratio of the normalized reads in the oxidized vs. unoxidized samples represents the fraction of a given miRNA that is methylated and recovered after oxidation. This analysis showed that after 48 h of HENMT1ᐃC-T6B expression, 92.6% of miRNAs were methylated ≥5%, which was sufficient for their enrichment after oxidation (Fig. 2B). At 48 h, the miRNAs detected after oxidation represent the top 98.2% of the cumulative miRNA reads in RKO cells, indicating that the physiologically

important miRNAs are readily captured by mime-seq 2.0. Importantly, expression of HENMT1ᐃC-T6B, and thus 2′O-methylation of miRNAs, did not affect miRNA levels compared to untransduced cells (Fig. 2C).

The induction timecourse allowed us to examine methylation dynamics. We estimated rates of methylation for every miRNA reliably detected in RKO cells (>0.01 normalized reads) as the increase in methylation between the 6 and 12 h timepoints, before

most miRNAs reached a methylation plateau (Dataset EV1). miRNAs appeared to be methylated with rates that varied 4–5-fold and that were not dependent on the relative abundance of the different miRNAs (Fig. 2D), but are possibly related to miRNA biogenesis and turnover dynamics (Reichholf et al, 2019). Importantly, despite these differences in methylation rates, the sequencing of methylated miRNAs at steady state maintained overall information on relative miRNA expression levels (Fig. 2E). Thus, mime-seq 2.0 is semi-quantitative. Altogether, these experiments indicate that HENMT1$^{\Delta C}$-T6B efficiently methylates the miRNome without substantial bias and should enable the implementation of mime-seq in mammals.

## HENMT1$^{\Delta C}$-T6B expression in mouse B cells enables specific miRNA retrieval without affecting differentiation

To implement mime-seq 2.0 in vivo, we generated the $R26^{\text{LSL-HenT6B/+}}$ mouse line by CRISPR/Cas9-mediated genome editing in 2-cell embryos (Gu et al, 2018). Upon Cre-mediated deletion of the loxP-Stop-loxP (LSL) cassette, the resulting $R26^{\text{HenT6B/+}}$ mice give rise to conditional expression of the HENMT1$^{\Delta C}$-T6B-P2A-eGFP gene from the ubiquitously expressed CAG promoter in the *Rosa26* locus (Figs. 3A and EV2). Hence, specific expression of HENMT1$^{\Delta C}$-T6B in different cell types can be achieved with appropriate cell type-specific Cre lines. Moreover, the simultaneous expression of GFP also allows for the visualization of HENMT1$^{\Delta C}$-T6B-expressing cells in their tissue context or for their isolation if desired. The murine hematopoietic system is ideal for validating the HENMT1$^{\Delta C}$-T6B-dependent mime-seq in vivo. First, because cells are more easily accessible than in other tissues, we can directly compare miRNAs from a purified cell population to a cell mixture, to evaluate the performance of mime-seq. Second, because the different cells of the lineage can be analyzed by flow cytometry, we could quantitatively assess the effect of HENMT1$^{\Delta C}$-T6B expression in different cell types. This is particularly important given that overexpression of a T6B peptide in the mouse was used as a competitive inhibitor to decrease miRNA function (La Rocca et al, 2021), and that inhibition of miRNA activity causes various defects (DeVeale et al, 2021).

We first expressed HENMT1$^{\Delta C}$-T6B specifically in B cells using the *Cd79a*-Cre line, which induces Cre activity in pro-B and all subsequent stages of B cell development (Hobeika et al, 2006). As shown by flow-cytometric analysis, pro-B, pre-B, recirculating and total B cells in the bone marrow as well as mature B cells in the spleen were minimally affected in *Cd79a*-Cre $R26^{\text{LSL-HenT6B/+}}$ and *Cd79a*-Cre $R26^{\text{LSL-HenT6B/LSL-HenT6B}}$ mice compared with control $R26^{\text{LSL-HenT6B/LSL-HenT6B}}$ mice (Figs. 3B,C and EV3A,B). Moreover, GFP and, by inference, HENMT1$^{\Delta C}$-T6B were expressed in splenic B cells, but not in T cells of *Cd79a*-Cre $R26^{\text{LSL-HenT6B/+}}$ and *Cd79a*-Cre $R26^{\text{LSL-HenT6B/LSL-HenT6B}}$ mice, although the expression was slightly variegated (Fig. EV3B). The presence of the different B cell types in *Cd79a*-Cre $R26^{\text{LSL-HenT6B/LSL-HenT6B}}$ mice is contrasted by the strong reduction of pre-B cells and absence of all subsequent B cell stages upon conditional Dicer deletion, leading to the loss of miRNAs in *Cd79a*-Cre *Dicer1*$^{\text{fl/fl}}$ mice (Koralov et al, 2008). We conclude that the expression of HENMT1$^{\Delta C}$-T6B minimally affects B cell development and is thus compatible with normal miRNA function.

To assess the efficiency and specificity of mime-seq in vivo, we isolated CD43$^-$ mature B cells from the spleen of *Cd79a*-Cre

$R26^{\text{LSL-HenT6B/+}}$, *Cd79a*-Cre $R26^{\text{LSL-HenT6B/LSL-HenT6B}}$, and control $R26^{\text{LSL-HenT6B/LSL-HenT6B}}$ mice by immunomagnetic depletion of non-B cells. We extracted total RNA, added a mix of methylated and unmethylated spike-ins (see Methods) and generated small RNA sequencing libraries with and without prior oxidation treatment. The methylated spike-ins are used for normalization as before, the unmethylated spike-ins provide a control to assess the extent of depletion of non-methylated species after the oxidation treatment. After normalization, we observed that libraries from unoxidized samples showed no significant differences between cells with or without methyltransferase expression, indicating that expression of HENMT1$^{\Delta C}$-T6B does not affect miRNA abundance (Fig. 3D left panel). Second, we observed that upon oxidation, miRNAs from the mature B cells of control $R26^{\text{LSL-HenT6B/LSL-HenT6B}}$ mice were depleted to levels comparable with unmethylated spike-ins, indicating that the oxidation treatment efficiently removes non-methylated miRNAs from the pool of sequenced miRNAs (Fig. EV3C). In contrast, HENMT1$^{\Delta C}$-T6B expressing B cells showed efficient recovery of miRNAs after oxidation, with the homozygous *Cd79a*-Cre $R26^{\text{LSL-HenT6B/LSL-HenT6B}}$ B cells showing higher levels of methylation/recovery after oxidation than heterozygous *Cd79a*-Cre $R26^{\text{LSL-HenT6B/+}}$ B cells (Figs. 3D,E and EV3C). Importantly, the relative abundance of miRNAs recovered after oxidation was highly similar to the relative abundance of miRNAs in B cells without methylation or oxidation treatment, again showing the semi-quantitative nature of mime-seq (Fig. 3D). In addition, miRNAs recovered after oxidation included all miRNA families known to be important for B cell development and homeostasis (Calderón et al, 2021) (Fig. 3F). As the CD43$^-$ mature B cells also contained a fraction of contaminating erythrocytes, we could demonstrate that miRNAs known to be absent from B cells, like the erythrocyte-specific miR-451a and miR-144 (Rasmussen et al, 2010), were fully depleted upon oxidation even if they were present at high levels in the starting sample (Fig. 3F). To summarize, mime-seq 2.0 enables specific detection of miRNAs from a cell of interest without strongly influencing miRNA expression and without causing detrimental effects that would be expected from a loss of miRNA function.

## Mime-seq 2.0 enables the specific enrichment of miRNAs from a rare cell population

As the main advantage of mime-seq is to facilitate miRNA profiling in rare cell populations, it is important to assess its sensitivity. To this end, we used the *Bhlha15*-Cre line for conditional activation of HENMT1$^{\Delta C}$-T6B expression in plasma cells (PCs), as the *Bhlha15* (Mist1) gene is specifically expressed in PCs within the hematopoietic system (Wöhner et al, 2022). Seven days after immunization with sheep red blood cells, GFP$^+$ PCs (CD138$^+$TACI$^+$) were isolated from the spleen of *Bhlha15*-Cre $R26^{\text{LSL-HenT6B/LSL-HenT6B}}$ mice by flow-cytometric sorting (Fig. EV4A). Purified PCs were then mixed at different ratios with wild-type C57BL/6 splenocytes, and these were used for miRNA sequencing before and after oxidation (Fig. 4A). The miRNA profiles of PCs (100%) and splenocytes (0.01% PCs) are highly similar with one miRNA being exclusively expressed in PCs, miR-148a (Fig. EV4B), which is necessary for PC differentiation and survival (Porstner et al, 2015). Highlighting the specificity and sensitivity of mime-seq, miR-148a was robustly detected in cell mixes containing 1% and 0.1% PCs, and, although the signal was overall lower, miR-148a was even detected above background in a population with only 0.01% PCs (Figs. 4B

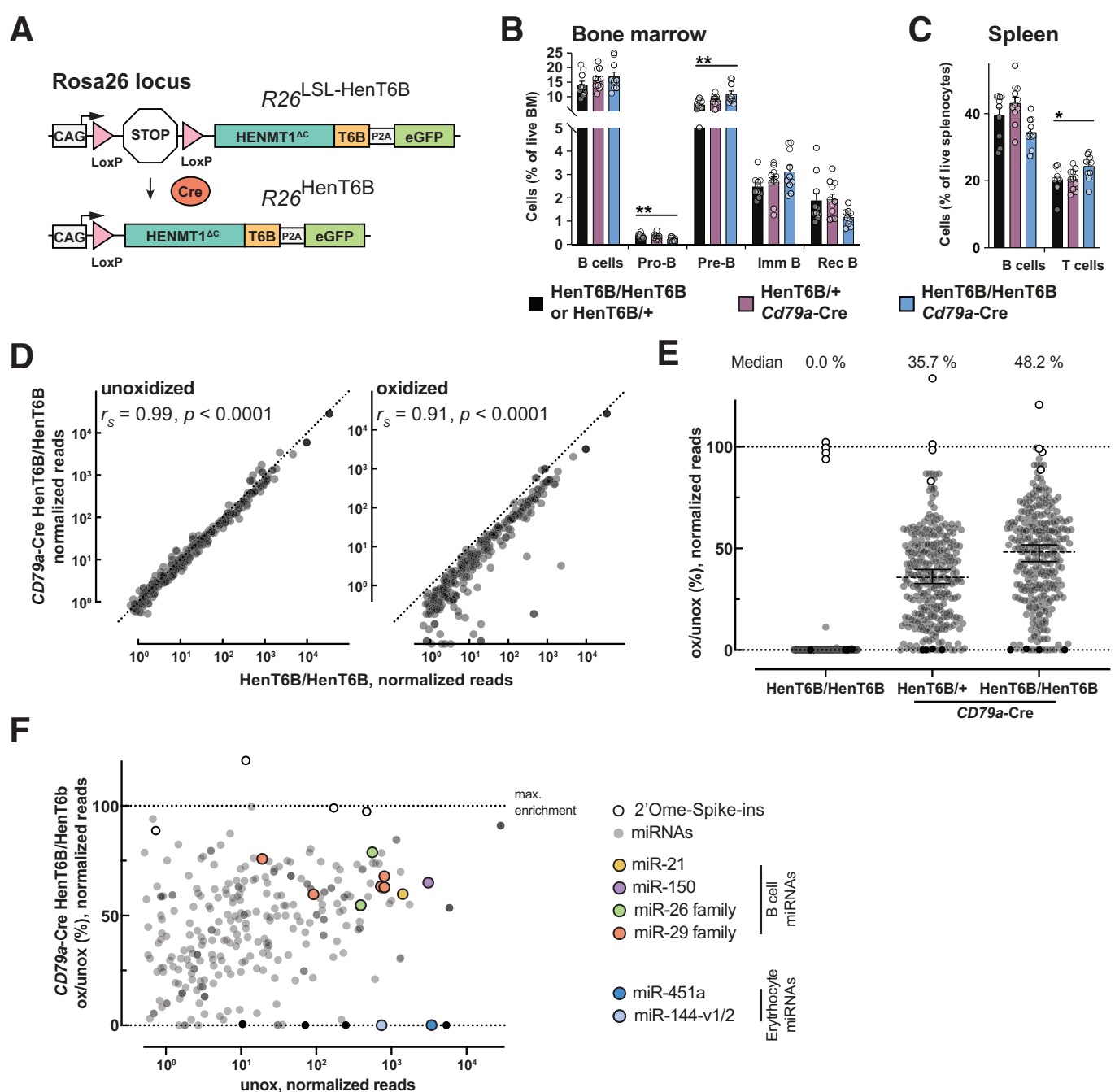

**Figure 3. Mime-seq 2.0 allows selective enrichment of mouse B cell miRNAs without affecting B cell differentiation.**

(A) Schematic diagram of the $R26^{LSL\text{-}HenT6B}$ and $R26^{HenT6B}$ alleles. The HENMT1$^{\Delta C}$-T6B-P2A-eGFP gene is expressed under the control of the CMV enhancer and chicken actin promoter (CAG) from the $R26^{HenT6B}$ allele upon Cre-mediated deletion of the loxP-Stop-loxP (LSL) cassette. The CAG promoter in the Rosa26 (R26) locus is permissive for ubiquitous expression in different tissues of the mouse. In all further figures, $R26^{LSL\text{-}HenT6B/+}$ and $R26^{LSL\text{-}HenT6B/LSL\text{-}HenT6B}$ are referred to as HenT6B/+ and HenT6B/HenT6B, respectively. (B, C) Frequencies of different B cell types in the bone marrow (B) and spleen (C) of 6–8-week-old Cd79a-Cre $R26^{LSL\text{-}HenT6B/+}$ (violet), Cd79a-Cre $R26^{LSL\text{-}HenT6B/LSL\text{-}HenT6B}$ (blue) and control $R26^{LSL\text{-}HenT6B/+}$ or $R26^{LSL\text{-}HenT6B/LSL\text{-}HenT6B}$ (black) mice were determined by flow-cytometric analysis, as shown in Fig. EV3A,B. The frequencies of total B, pro-B, pre-B, immature (imm) B, recirculating (rec) B cells and total T cells are shown as mean values of three independent experiments with SEM, based on the analysis of at least 10 mice per genotype. Statistical data (B, C) were analyzed by one-way ANOVA with Dunnett's multiple comparisons test: *$P < 0.05$, **$P < 0.01$. The flow-cytometric definition of the different B cell types is described in Methods. (D) Spike-in-normalized small RNA libraries from the same genotypes. Expression of HENMT1$^{\Delta C}$-T6B does not affect miRNA abundance (unoxidized), and the recovered miRNAs after oxidation largely maintain their relative abundance (oxidized) ($r_S$=Spearman correlation coefficient). (E) Percentage of recovered reads (ox/unox) are shown. Median level of methylation for every genotype is indicated above and as lines with 95% CI. $N = 292$ miRNAs per timepoint. Methylated spike-ins shown in white, non-methylated spike-ins shown in black. Sequencing libraries were prepared from one biological replicate each. (F) Methylation and recovery efficiency as a function of overall abundance in the starting sample. Note enrichment of known B cell miRNAs and depletion of the very abundant miR-451a and miR-144, two miRNAs from contaminating erythrocyte that do not express the methyltransferase. Source data are available online for this figure.

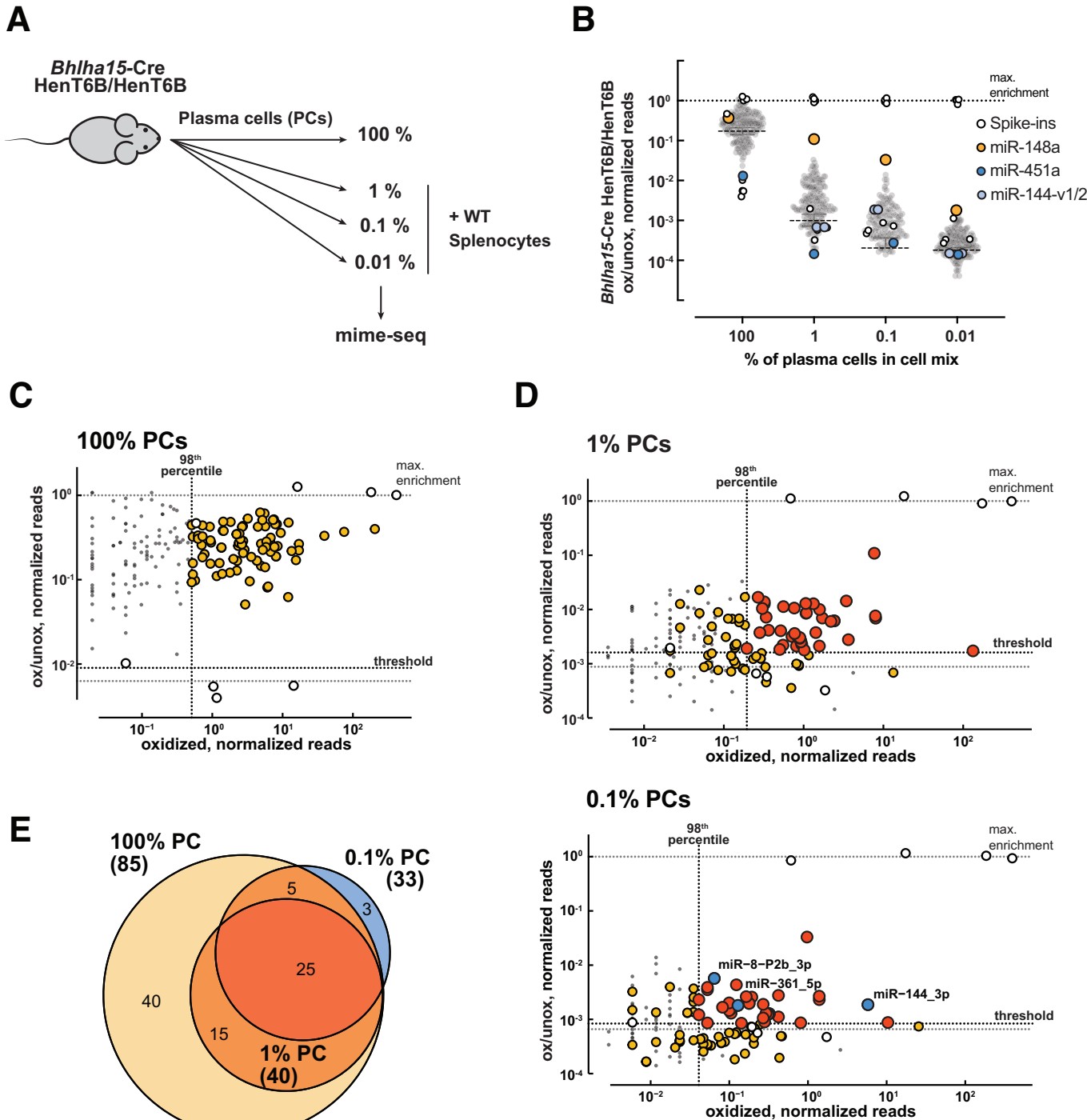

and EV4C). As expected, miRNAs that are expressed in both PCs and splenocytes were enriched to a lower degree than a highly cell-specific miRNA like miR-148a (Fig. EV4C; Dataset EV2), but we could still use the full miRNomes to assess specificity and sensitivity more broadly. We first determined a reference PC miRNome composed of 85 miRNAs that account for the top 98% of cumulative reads in the oxidized purified PC population (Fig. 4C; Dataset EV2). We did not use the unoxidized sample because it still contained erythrocytes, resulting in the presence of specific

erythrocyte miRNAs that are eliminated after oxidation. We then asked how many of these miRNAs are robustly detected (in the 98th top percentile) after oxidation in the various cell mixes, and show significant enrichment after oxidation (ox/unox ratio above a cutoff set as the average + standard deviation of the ox/unox ratio of the unmethylated spike-ins, which represent the maximum expected depletion) (Figs. 4D, E and EV4E). With these cutoffs, we detected 0 false positives (FPs) in the 1% PC mix and 3 FPs in the 0.1% mix. Moreover, in the 1% PC mix, we recovered 40 of all 85

◄  **Figure 4.   HENMT1$^{\Delta C}$-T6B mime-seq recovers cell-specific miRNAs in vivo.**

(A) Schematic overview of the mixing experiment to test the sensitivity of mime-seq. Plasma cells (PCs) were isolated from the spleen of immunized *Bhlha15*-Cre *R26*$^{HenT6B/HenT6B}$ mice by flow-cytometric sorting (Fig. EV4A) and were then mixed at the indicated ratios with total splenocytes of wild-type (WT) C57BL/6 mice. Mime-seq was performed with the different cell mixes. (B) Total RNA extracted from *Bhlha15*-Cre *R26*$^{LSL-HenT6B/ LSL-HenT6B}$ PCs mixed at indicated ratios with B6 splenocytes. A mix of methylated and unmethylated spike-ins was added and small RNA sequencing libraries prepared from RNA with or without prior oxidation. Reads are normalized to methylated spike-ins. Shown are fractions recovered for all sequenced miRNAs with normalized reads >0.5. Median level of methylation for every genotype is shown as lines with 95% CI. $N = 295$ miRNAs per timepoint. Sequencing libraries were prepared from one biological replicate. (C, D) Identification of miRNAs confidently expressed in PCs. Black dotted lines indicate the thresholds use to define miRNAs expressed in PCs. We used an abundance cutoff (vertical line) for miRNAs that are in the top 98$^{th}$ percentile of cumulative reads, and an enrichment after oxidation cutoff (horizontal line) calculated as the average + standard deviation of the ox/unox ratio of the four unmethylated spike-ins. Shown are fractions recovered for all sequenced miRNAs with normalized reads >0.5 (PC mixing ratios indicated above). Reads originating from isomiRs are summed. (C) miRNAs identified in 100% PC (yellow). (D) miRNAs identified in 1% or 0.1% PC (orange). False positives (blue) are defined as miRNAs that pass the set thresholds but are not identified in the 98$^{th}$ top percentile of the 100% PC sample. (E) Venn-diagram showing overlap between different samples. Colors as described above.

PC miRNAs, but most importantly we retrieved 5 of the top 5, 9 of the top 10, 22 of the top 25, and 36 of the top 50 PC miRNAs. In the 0.1% mix, we still detected 26/50 of the top 50 miRNAs (Figs. 4D and EV4E). Our results suggest that for broadly expressed miRNAs, mime-seq retrieves the most likely physiologically relevant miRNAs (based on abundance) in cells that are present in as little as 1 in 100. Most importantly, mime-seq is best suited to reliably uncover the miRNAs that are specifically expressed in a particular cell type within a mixed cell population, even when that cell type is present in as little as 1 in 1000.

## Discussion

Here we presented mime-seq 2.0, a method for in vivo profiling of miRNAs from specific cell types of choice in the mouse. Mime-seq 2.0 relies on transgenic expression of an engineered methyltransferase that "marks" miRNAs in a cell type of choice. Cell-specific miRNAs can be retrieved from total RNA from a complex cell mix by a simple oxidation treatment that enables selective cloning and sequencing of the 2′O-methylated miRNAs. The addition of methylated and unmethylated spike-ins allows for normalization, additional quantification and provides a control for the oxidation treatment. Because methylation is largely unbiased across the miRNome, it allows for semi-quantitative profiling of miRNAs from the cells expressing the methyltransferase.

When we developed the first version of mime-seq, we employed the plant enzyme At-HEN1 (Alberti et al, 2018). This enzyme, which recognizes the short RNA duplexes produced upon Dicer cleavage, efficiently methylated miRNAs in *C. elegans* and *Drosophila*. However, here we report that At-HEN1 was unable to methylate miRNAs in mouse or human cell lines. This is consistent with the fact that in mammals, Dicer and Argonaute form part of a miRNA loading complex in which the product of Dicer cleavage has been proposed to be "handed over" to Argonaute (Maniataki and Mourelatos, 2005; Wang et al, 2009; Sasaki and Shimizu, 2007). The stark difference in methylation activity by At-HEN1 in vivo in worms and flies versus mammals indicates that loading mechanisms are fundamentally different among these species. To overcome the experimental challenge imposed by this difference, we designed a chimeric enzyme based on the methyltransferase domain of mouse HENMT1 and imparted it with specificity towards miRNAs by fusing it to the T6B peptide that interacts with all four mammalian Argonaute proteins into

which miRNAs are loaded for function (Hauptmann et al, 2015). Notably, only HENMT1$^{\Delta C}$-T6B, but not full-length HENMT1 or its methyltransferase domain alone, was able to methylate miRNAs. This suggests that miRNA methylation by HENMT1$^{\Delta C}$-T6B occurs after loading into Argonaute.

For cell-specific methylation in the mouse, we generated a conditional *R26*$^{LSL-HenT6B/LSL-HenT6B}$ mouse line that can be crossed with a Cre driver of choice to ultimately induce cell-specific miRNA methylation. This approach leverages the multitude of Cre recombinase drivers available for the mouse, most of which have been well characterized (i.e., JAX Cre repository). While good description of driver expression is essential for the interpretation of mime-seq results, the eGFP in the same transgene allows for rapid validation of HENMT1$^{\Delta C}$-T6B expression patterns. If necessary, coarse dissection of the tissue of choice could be used to further enhance specificity of a given driver. The *R26*$^{LSL-HenT6B/LSL-HenT6B}$ mice are readily available, and HENMT1$^{\Delta C}$-T6B mice can be generated by a simple crossing scheme. Notably, double heterozygous mice provide sufficient methylation for the implementation of mime-seq, although homozygous expression of HENMT1$^{\Delta C}$-T6B can be used to increase the extent of methylation, if necessary (Figs. 3E and EV3C).

When testing mime-seq 2.0 in B cell types of the mouse hematopoietic system, we found no indication of inhibition of miRNA function by HENMT1$^{\Delta C}$-T6B. Overexpression of a T6B-YFP fusion protein was, however, previously shown to inhibit miRNA function by competing for binding of endogenous TNRC6 to Argonaute, although this did not affect the miRNA repertoire (La Rocca et al, 2021). Within the hematopoietic system, T6B-YFP overexpression under the control of a doxycycline-inducible promoter in the *Col1a1* locus resulted in an increase of pro-B cells and a decrease of pre-B, immature and mature B cells in the bone marrow (La Rocca et al, 2021). In contrast, we observed similar frequencies of all B cell types in the bone marrow upon B cell-specific expression of HENMT1$^{\Delta C}$-T6B from the *Rosa26* locus. Given that T6B acts as a competitive inhibitor, this discrepancy might be explained by differences in expression level, or binding affinity. In our study, the expression of HENMT1$^{\Delta C}$-T6B seems to be sufficient to induce the necessary levels of miRNA methylation without affecting the generation or viability of the cells expressing the enzyme in contrast to the overexpression of T6B-YFP protein.

One of the main advantages of mime-seq is that it adds just one extra step (the oxidation treatment) to the standard ligation-based

small RNA sequencing protocol. This compares favorably with the technical involvement of Argonaute immunoprecipitation-based approaches, which, similar to mime-seq, also require cell-specific expression of a transgene (Kudlow et al, 2012; He et al, 2012). Mime-seq also compares favorably with approaches needing isolation of cells of interest by FACS or microdissection. These extra steps are not only technically involved, but they also subject cells to stress that may affect their miRNA expression profile.

The miRNome of a given cell type is composed of miRNAs that are shared with other cells, as well as other miRNAs that are cell type-specific. Mime-seq will retrieve a large fraction of the shared miRNAs, in particular those that are most abundant. However, miRNAs that are more abundant in other cells in the mix than in the cell type of interest will display low oxidized (specific)/unoxidized (total) read ratios. This is also a caveat of immunoprecipitation-based approaches as the technical background associated with this purification must be accounted for by assessing the ratio of IP (specific)/input (total) read ratios. Mime-seq performs best at identifying miRNAs that are specifically expressed in a cell type of interest but not in other cells in the sample, as this results in very high oxidized (specific)/unoxidized (total) read ratios. In those cases, mime-seq can uncover a miRNA that is specifically expressed in 1/1000 cells (Fig. 4B). Overall, mime-seq will reveal the miRNAs that are most likely functionally relevant in the cell type of interest, by identifying those that are most abundant and those that are highly cell type-specific. Given its technical advantages, we expect that mime-seq 2.0 will increase the resolution of miRNA profiling efforts to gain the necessary cellular level view to understand miRNA function.

# Methods

### Reagents and tools table

| Reagent/resource | Reference or source | Identifier or catalog number |
|---|---|---|
| **Experimental models** | | |
| HEK293T cell line (*H. sapiens*) | Takara | 632180 |
| RKO cells (*H. sapiens*) | ATCC | CRL-2577 |
| MEF cells (*M. musculus*) | Laboratory of Gijs Versteeg, University of Vienna | N/A |
| Mouse: R26(LSL-miR17-92) | Laboratory of Klaus Rajewsky, Max Delbruck Centre for Molecular Medicine (Xiao et al, 2008) | Stock#008517; RRID: IMSR_JAX:008517 |
| Mouse: Bhlha15-Cre | Laboratory of Stephen F. Konieczny, Purdue University | N/A |
| Mouse: Cd79a-Cre | The Jackson Laboratory | Stock#020505, RRID: IMSR_JAX:020505 |
| **Recombinant DNA** | | |
| Plasmids | | Table EV1 |
| **Antibodies** | | |
| Mouse anti-myc | Merck Millipore | 05-724 |

| Reagent/resource | Reference or source | Identifier or catalog number |
|---|---|---|
| Rabbit anti-At-HEN1 | Agrisera | AS15 3095 |
| Rabbit anti-actin | Sigma | A2066 |
| anti-rabbit HRP | Cell Signaling Technology | 7074 |
| anti-mouse HRP | Cell Signaling Technology | 7076 |
| anti-FLAG | Sigma | F1804 |
| Kit-PeCy7 (ACK) | Biolegend | b135112 |
| Kit-PeCy7 (2B8) | Invitrogen | 25-1171-82 |
| CD25-PE (PC61.5) | BD Bioscience | 553866 |
| IgM-PerCP-eFlour710 (II/41) | Invitrogen | 46-5790-82 |
| B220-BV785 (RA3-6B2) | BD Bioscience | 563894 |
| IgD-APC (11-26C) | Invitrogen | 17-5993-82 |
| CD4-BV510 (GK1.5) | BD Bioscience | 569249 |
| CD8a-BV605 (53-67) | BD Bioscience | 563152 |
| CD21/CD35-PE (7G6) | BD Bioscience | 562756 |
| CD19-BV421 (1D3) | BD Bioscience | 562701 |
| TCRb-APC-eFlour780 (H57-597) | Invitrogen | 47-5961-80 |
| CD138-APC (281-2) | BD Bioscience | 561705 |
| TACI-PE (8F10) | BioLegend | 133403 |
| **Oligonucleotides and sequence-based reagents** | | |
| DNA/RNA oligo sequences | | Table EV1 |
| **Chemicals, enzymes, and other reagents** | | |
| Gibson Assembly Master Mix | New England Biolabs | E2611L |
| DMEM | Corning | 10-013-CV |
| RPMI | Corning | 10-040-CV |
| FBS | Avantor | 89510186 |
| l-glutamine | Gibco | 25030081 |
| Sodium Pyruvate | Sigma | S8636 |
| HEPES | Sigma | H4034-500G |
| Polyethylenimine, Linear, MW 25000 | Polysciences | 23966 |
| Doxycycline | Sigma | D9891-1G |
| Opti-MEM | Gibco | 31985062 |
| PBS | Corning | 21-040-CV |
| Trizol | Invitrogen | 15596026 |
| Polybrene | Sigma | TR-1003-G |
| 0.45 μm filters | Sarstedt | 83.1826 |
| Blasticidin | InvivoGen | ant-bl-05 |
| CD43-MicroBeads | Miltenyi Biotec | #130-049-801 |
| Pierce BCA Protein Assay | Thermo Scientific | 23227 |

| Reagent/resource | Reference or source | Identifier or catalog number |
|---|---|---|
| 4–20% Mini-PROTEAN® TGX™ Precast Protein Gels | BioRad | 4561093 |
| Nitrocellulose membrane | BioRad | |
| Clarity Western ECL | BioRad | 1705060 |
| 2-propanol | Fisher Chemical | A416-1 |
| Chloroform | Fisher Chemical | C298-500 |
| Glycogen | Sigma | G1767-1VL |
| EtOH abs. | Avantor | 20821.321 |
| $H_2O$, DEPC-treated | Invitrogen | AM9920 |
| $H_2O$, Rnase free | Invitrogen | AM9935 |
| Qubit RNA BR Assay Kit | Invitrogen | Q10210 |
| Ammonium Persulfate | Sigma | A3678 |
| TEMED | Sigma | T9281 |
| Sodium Periodate | Sigma | 311448-5G |
| Sodium Acetate (3 M) ph 5.5 | Invitrogen | AM9740 |
| NaOH | Sigma | S5881 |
| Boric acid | Sigma | B6768 |
| Sodium tetraborate | Sigma | 311448-5G |
| Sodium chloride | Sigma | 106404 |
| SequaGel UreaGel System | National Diagnostics | EC-833 |
| Gel Loading Buffer II | Invitrogen | AM8547 |
| Hybond-NX membrane | Cytiva | RPN303T |
| Methylimidazole | Sigma | M50834 |
| EDC | Sigma | E7750-25G |
| EDTA (0.5 M) pH 8 | Invitrogen | 15575020 |
| Sodium Phosphate dibasic heptahydrate | Sigma | 431478 |
| Phosphoric Acid | Avantor | 02-003-602 |
| T4 Polynucleotide Kinase | New England Biolabs | M0201L |
| γ-$^{32}$P-ATP | Perkin Elmer | BLU002Z250UC |
| MicroSpin G-25 Columns | Cytiva | 27532501 |
| Storage Phosphor Screen | Cytiva | 28956474 |
| Protein G Dynabeads | Invitrogen | 10003D |
| K227Q truncated T4 RNA ligase 2 | New England Biolabs | M0351L |
| PEG8000 50% | New England Biolabs | B1004A |
| SYBR Gold | Invitrogen | S11494 |
| TBE buffer, 10x | Quality Biological | 351-001-131 |
| SSC buffer, 20x | Sigma | S6639-1L |

| Reagent/resource | Reference or source | Identifier or catalog number |
|---|---|---|
| T4 RNA ligase 1 | New England Biolabs | M0204L |
| Superscript II or III reverse transcriptase | Invitrogen | 18064014/ 18080044 |
| RNaseOUT | Invitrogen | 10777019 |
| ExoSAP-IT | Applied Biosystems | 78201 |
| Kapa HiFi HotStart Library Amp kit | Roche | KK2612 |
| Zymoclean Gel DNA recovery kit | Zymo | D4008 |
| Low Range Agarose | BioRad | 1613107 |
| IGEPAL CA-630 | Sigma | I3021 |
| Sodium Deoxycholate | Sigma | D6750 |
| Sodium Dodecyl Sulfate | Sigma | L3771 |
| Sodium Dodecyl Sulfate 20% solution | Sigma | 05030 |
| Tris | Thermo Scientific | 75825 |
| Glycerol | Sigma | 15523 |
| Bromophenol Blue | BioRad | 1610404 |
| 2-Mercaptoethanol (BME) | Sigma | M-6250 |
| Tween | Sigma | P9416 |
| Magnesium Chloride | Avantor | 7791-18-6 |
| **Software** | | |
| FlowJo 10.10 (Treestar) | https://www.flowjo.com | |
| GraphPad Prism 10.1.1 | https://www.graphpad.com | |
| R Studio v2022.12.0 | https://posit.co | |
| Adobe Illustrator v25.4.1 | https://www.adobe.com | |
| **Other** | | |
| LSRFortessa | BD Biosciences | |
| FACSAria III | BD Biosciences | |
| Aurora | Cytek | |
| Trans-Blot SD | BioRad | |
| Typhoon phosphorimager | Cytiva | |
| HiSeq V4 | Illumina | |
| NovaSeq | Illumina | |
| Qubit flex fluorometer | Invitrogen | |
| Blue light transilluminator | | |
| ChemiDoc | BioRad | |

## Vectors

All HENMT1 and At-HEN1 constructs were cloned into a modified lentiviral pLX303 vector under a SFFV promoter. Sequences for HENMT1-T6B and codon-optimized At-HEN1 were ordered as

gene blocks, modified constructs shown in Fig. 1A were generated using Gibson assembly or side-directed mutagenesis. To generate inducible expression vectors, SFFV promoter was replaced with a TRE3 promoter by Gibson assembly. All constructs were also cloned into a minimal pCS2 vector under a CMV IE94 promoter for transfection. All plasmids used in this study are provided in Dataset EV3. Lentiviral HENMT1$^{\Delta C}$-T6B plasmid will be made available through Addgene.

## Cell maintenance

Cell lines were maintained in DMEM (HEK293T, MEF) or RPMI (RKO) supplemented with 10% FBS (Avantor, 89510186), 4 mM l-glutamine (Gibco, 25030081), 1 mM Sodium Pyruvate (Sigma, S8636) and 25 mM HEPES (Sigma, H4034-500G) at 37 °C, 5% $CO_2$. Cell lines have been tested regularly for mycoplasma contamination.

## Transfection

$6 \times 10^5$ cells per well were seeded the day before transfection in a 6-well plate. 1 μg plasmid DNA was transfected with polyethylenimine (PEI, Polysciences, 23966) at a ratio of 1:3 (μg DNA:μg PEI) in 200 μl Opti-MEM (Gibco, 31985070). Media was replaced after 24 h. After 48 h cells were washed with ice-cold PBS and harvested either in RIPA buffer (150 mM NaCl, 1% IGEPAL CA-630, 0.5% Sodium deoxycholate, 0.1% SDS, 50 mM Tris pH 8) for Western blot experiments or in TRIzol (Invitrogen, 15596026) for RNA extraction. Samples were stored at −80 °C.

## Lentivirus production and target cell transductions

Commercial LentiX (Takara, 632180) at 70% confluency were transfected with a mix of Eco envelope plasmid (Cell Biolabs, 320026), pCMVR8.74 (Addgene, 22036) and transfer plasmid of interest. Supernatant, containing VLPs, was harvested after 48 h, 56 h and 72 h, filtered (0.45 μm) and used for target cell transduction in the presence of 4 μg/ml polybrene (Sigma). Blasticidin (InvivoGen, ant-bl-05) was used at a concentration of 5 μg/ml for selection of integrated constructs.

## Mouse-related methods

The following mice were maintained on the C57BL/6 genetic background: Cd79aCre/+ mice (Hobeika et al, 2006), homozygous miR-17-92 transgenic mice (Xiao et al, 2008) (here referred to as R26$^{LSL-miR-17-92/LSL-miR17-92}$ mice), and Bhlha15Cre/+ mice (generated in the lab of Stephen F. Konieczny, Purdue University, West Lafayette, USA). Cd79aCre/+ and Bhlha15Cre/+ mice are referred here as *Cd79a*-Cre and *Bhlha15*-Cre mice, respectively. Experimental and control mice were co-housed under standard pathogen-free conditions at a temperature of 22 °C and 55% humidity with a day cycle of 14 h light and 10 h dark and with unrestricted access to food and water. Mice were euthanized by carbon dioxide inhalation. Both female and male mice were used at a similar ratio in this study. All animal experiments were carried out according to valid project licenses, which were approved and regularly controlled by the Austrian Veterinary Authorities.

## Generation of the R26$^{LSL-HenT6B}$ allele

For generating the *R26*$^{LSL-HenT6B}$ allele, the HENMT1$^{\Delta C}$-T6B cDNA was first cloned in the CAG-STOP-eGFP-Rosa26 TV plasmid (Addgene, 15912) by replacing the IRES sequence with the HENMT1$^{\Delta C}$-T6B cDNA fused in frame via a P2A peptide to the eGFP coding sequence to generate the CAG-STOP-HENMT1$^{\Delta C}$-T6B-eGFP-Rosa26 plasmid (Fig. 2A). A 3709-bp long DNA fragment was PCR-amplified from the CAG-STOP-HENMT1$^{\Delta C}$-T6B-eGFP-Rosa26 plasmid with an upstream primer (5′-CTGG CACTTCTTGGTTTTCC-3′) and downstream primer (5′-GCTGC ATAAAACCCCAGATG-3′) (Fig. EV2). The *R26*$^{LSL-HenT6B}$ allele was generated by CRISPR/Cas9-mediated genome editing in mouse 2-cell embryos (2C-HR-CRISPR) (Gu et al, 2018). For this, 2-cell embryos of the *R26*$^{LSL-miR-17-92/LSL-miR17-19}$ genotype (C57BL/6) were injected with Cas9 protein, two appropriate sgRNAs (linked to the scaffold tracrRNA) and the double-stranded 3709-bp DNA repair template to generate the *R26*$^{LSL-HenT6B}$ allele (Fig. EV2). Correct targeting of the *R26*$^{LSL-HenT6B}$ allele was verified by DNA sequencing of respective PCR fragments. The *R26*$^{LSL-HenT6B}$ allele was genotyped by amplification of a 334-bp PCR fragment from the HENMT1$^{\Delta C}$ insert with primer 1 (5′-ATGCCAAGCTCCTAAAGCTG-3′) and primer 2 (5′-GGGTTGAATTCAGCATTTGG-3′). In a separate PCR, the wild-type *R26*$^+$ allele was genotyped by amplification of a 170-bp PCR fragment with primer 3 (5′-CTCTTCCCTCGTGATCTGCAACTCC-3′) and primer 4 (5′-TCCCGACAAAACCGAAAAT-3′).

## Definition of the cell types by flow cytometry

The different hematopoietic cell types of the mouse were identified by flow cytometry as follows: pro-B (B220$^+$CD19$^+$Kit$^+$CD25$^-$IgM$^-$IgD$^-$), pre-B (B220$^+$CD19$^+$Kit$^-$CD25$^+$IgM$^-$IgD$^-$), immature B (B220$^+$CD19$^+$IgM$^{hi}$IgD$^-$), recirculating B (B220$^+$CD19$^+$IgM$^+$IgD$^{hi}$), plasma cells (CD138$^+$TACI$^+$), total B cells (B220$^+$CD19$^+$) and total T cells (CD3$^+$TCRβ$^+$). Flow-cytometric analysis was performed on the LSRFortessa (BD Biosciences) machine and flow-cytometric sorting of plasma cells on the FACSAria III (BD Biosciences) machine. FlowJo Software (Treestar) was used for data analysis. CD43$^-$ B cells were enriched from the spleen by immunomagnetic depletion of non-B cells with CD43-MicroBeads (Miltenyi Biotec). The following monoclonal antibodies were used for flow-cytometric analysis of mouse lymphoid organs from 4- to 12-week-old mice: B220/CD45R (RA3-6B2), CD3 (17A2), CD4 (GK1.5), CD8a (53-67), CD11b/Mac1 (M1/70), CD19 (1D3), CD21/CD35 (7G6), CD23 (B3B4), CD25 (PC61.5), CD117/Kit (ACK2), CD138 (281-2), CD267/TACI (8F10), IgD (11-26C), IgM (II/41), and TCRβ (H57-597) antibodies.

## Immunization

Sheep red blood cells were washed in PBS and resuspended at 10$^9$ cells/ml followed by intraperitoneal injection of 100 ml into an adult mouse. Plasma cells from *Bhlha15*-Cre *R26*$^{LSL-HenT6B/LSL-HenT6B}$ mice were isolated by flow-cytometric sorting at day 7 after immunization.

## Western blot

Cells were lysed in RIPA buffer. Protein concentration was determined by BCA (Thermo Scientific, 23227). After addition of

6x loading dye (200 mM Tris pH 6.8, 10% SDS, 60% Glycerol, 0.036% Bromophenol Blue, 5% BME), 15–30 µg of protein were denatured 5 min at 95 °C, separated on a 4–20% gradient gel (BioRad, 4561093) and transferred to a nitrocellulose membrane (BioRad). After blocking for 30 min (5% milk in Tris-buffered saline, 0.1% Tween20 - TBST), membrane with primary antibody was incubated overnight at 4 °C. The next day membranes were washed three times with TBST and incubated with HRP-conjugated secondary antibodies (5% milk in TBST) for 1 h at room temperature (RT). Membranes were washed three times with TBST and imaged with Clarity Western ECL (BioRad, 1705060). All blots were imaged with a ChemiDoc (BioRad) or Odyssey XF (LI-COR) imager. For detection of Myc-tagged HENMT1 constructs and Myc-At-HEN1 mouse monoclonal anti-Myc tag antibody, clone 4A6 (Merck Millipore, 05-724) at 1:2000–1:5000 dilution and rabbit anti-At-HEN1 antibody (Agrisera, AS15 3095) at 1:1000 dilution were used. For detection of Actin rabbit anti-Actin antibody (Sigma, A2066) was used at 1:2000 dilution. Secondary HRP-coupled antibodies, anti-rabbit IgG HRP-linked antibody (Cell Signaling Technology, 7074) and anti-mouse IgG HRP-linked antibody (Cell Signaling Technology, 7076) were used at 1:2000 dilution.

## RNA extraction and preparation

Samples were lysed in TRIzol. For frozen cell pellets, samples were thawed on ice before addition of 1 ml TRIzol. 200 µl chloroform (Fisher Chemical, C298-500) was added per 1 ml of TRIzol, samples vortexed and incubated for 3 min at RT. Samples were then centrifuged for 15 min at 4 °C (12,000 × $g$) to promote separation of organic (bottom) and aqueous phase (top). Clear phase on top was transferred to a new 1.5 ml tube and 1 µl glycogen (20 mg/ml, Sigma, missing) added. RNA was precipitated by addition of 1 volume isopropanol (Fisher Chemical, A416-1). Samples were vortexed, incubated 5 min at RT and pelleted by centrifugation for 10 min at 4 °C (12,000 × $g$). Pellets were washed with 80% ice-cold EtOH. For resuspension of RNA, pellets were centrifuged and air dried for 5 min before addition of RNase-free $H_2O$ (Ambion). RNA was quantified using the Qubit BR kit (Invitrogen, Q10210). For northern hybridization experiments, 5–15 µg RNA was subjected to oxidation followed by β-elimination per lane. For small RNA library preparation from mouse samples, when possible >3 µg RNA was mixed with methylated (Spike-ins mX2, mX3, mX6, and mX8) and unmethylated (Spike-ins X1, X3, X5, and X7) spike-ins (Lutzmayer et al, 2017) (sequences and concentrations are provided in Dataset EV3) before sample was split in half for oxidation with and without $NaIO_4$. Where indicated, libraries were generated from the maximum amount of RNA obtained, e.g., from sorted plasma cells we only retrieved 340 ng of RNA and still were able to produce good quality libraries. For samples obtained from human RKO cells, only methylated spike-ins were used (Spike-ins mX1-mX8).

## Northern blot

### Oxidation

For a typical 20 µl reaction, 5–15 µg RNA were used as input with or without 2 µl $NaIO_4$ (50 mM) and 4 µl 5x borate buffer (300 mM boric acid/borax, pH 8.6). Reactions were carried out at RT for 30 min.

### β-elimination

1 µl NaOH (1 M) was added for a final concentration of 50 mM and incubated at 45 °C for 90 min. Reactions were filled up with $H_2O$ to 300 µl with 300 mM NaOAc (pH 5.2, Ambion).

### Northern blotting

Samples were precipitated using 900 µl EtOH using glycogen as carrier and incubated for 1.5 h at −20 °C before pelleting for 30 min at 4 °C with 20,000 × $g$. After a wash step using 80% EtOH sample pellets were resuspended in 10 µl of formamide loading dye (Invitrogen, AM8547) and loaded on a 15% urea PAGE (National Diagnostics, EC-833). Northern blots were performed as previously described (Ameres et al, 2010) with minor modifications. Using bromophenol blue and xylene cyanole as reference, the gel was cut in two. After semidry transfer (BioRad) onto Hybond NX membranes (Cytiva, RPN303T), the lower part was used for hybridization and visualization of miRNAs, the upper part for the tRNA used as loading control and as a control that the oxidation reaction proceeded to completion. RNA was UV crosslinked three times with 120 mJ/cm$^2$ each, followed by chemical crosslinking using methylimidazole/EDC (Pall and Hamilton, 2008). Membranes were prehybridized with Church buffer (1 mM EDTA, 0.5 M $Na_2HPO_4/NaH_2PO_4$, 7% SDS) for 1 h at 65 °C in a hybridization oven. Probes were synthesized as ssDNA oligos and 5′ $^{32}$P-radiolabeled with T4 polynucleotide-kinase (NEB) and γ-$^{32}$P-ATP (6000 Ci/mmol, Perkin Elmer). All probe sequences used in this study are provided in Dataset EV3. Unincorporated nucleotides were removed by G25 column purification (Cytiva, 27532501). 25 pmoles radioactively labeled probes were added. Temperature was lowered to 37 °C and membranes incubated overnight. Membranes were washed at 37 °C three times with 1x SSC + 0.1% SDS for 10 min before exposure to a storage phosphor screen (Cytiva). All imaging was performed with a Typhoon phosphorimager (Cytiva).

## In vitro methylation assay

In vitro methylation assays were performed using immunopurified Myc-At-HEN1 from transiently transfected HEK293T cells or FLAG-Myc-At-HEN1 from stably expressing S2 cells. Immunoprecipitations were performed as previously described (Reimão-Pinto et al, 2015) using Protein G Dynabeads (Invitrogen) coupled to Myc 4A6 antibody (Merck Millipore, 05-724) or FLAG M2 monoclonal antibody (Sigma, F1804). All RNA substrate sequences can be found in Dataset EV3. For miRNA substrate preparation, guide strand was 5′ $^{32}$P-radiolabeled using T4 polynucleotide-kinase and unincorporated nucleotides removed by G25 column purification. After PAGE purification, labeled guide strands were annealed to 5′ phosphorylated miR* strands. Purified methyltransferases were incubated with labeled miRNA substrates in standard RNAi reactions containing S-adenosylmethionine. Following phenol/chloroform extraction, RNA was subjected to oxidation and β-elimination and run on a 15% denaturing PAGE. Dried gels were exposed to a storage phosphorscreen and imaged using a phosphorimager.

## Small RNA libraries

A detailed step-by-step version of the protocol is maintained at protocols.io: https://doi.org/10.17504/protocols.io.rm7vzxzzxgx1/v1.

## Sample preparation

### Oxidation

Performed as described above. For small RNA libraries, no β-elimination was performed. After oxidation, samples were purified by EtOH precipitation as described and resuspended in 6 μl H₂O.

### 3′ adapter ligation and purification

Barcoded 3′ adapters (see Dataset EV3) were ligated using K227Q truncated T4 RNA ligase 2 (NEB, M0351L) at a concentration of 0.5 μM adapter and in the presence of 25% PEG8000 containing a homemade 10x ligation buffer (0.5 M Tris pH 7.8, 0.1 M MgCl₂, 0.1 M DTT) overnight at 4 °C. As reference for later PAGE purification, independent ligation reactions were also set up for an 18-mer and 30-mer ssRNA oligos. 3′ adapter ligated small RNAs were loaded in formamide loading buffer on a 15% urea PAGE. After visualizing RNA with SYBR Gold (1:7500; Invitrogen, S11494) in 0.5x TBE, gel fragments spanning the ligated 18-mer and 30-mer were cut out. RNA was eluted from gel pieces in 800 μl gel elution buffer (0.3 M NaCl, 0.1% SDS) rotating overnight. To concentrate samples, RNA was precipitated by addition of 2.5–3x volumes of ice-cold EtOH for 1 h at −20 °C, after centrifugation for 30 min at 4 °C, RNA pellet was washed using 80% ice-cold EtOH and eluted in 6 μl H₂O.

### 5′ adapter ligation and purification

1 μl of 5′ adapter (10 μM) was added and incubated for 5 min at 65 °C before addition of T4 ligase buffer (NEB), final 25% PEG8000 and T4 RNA ligase 1 (NEB, M0204L). The ligation reaction was incubated overnight at 4 °C. RNA Clean & Concentrator kit (Zymo, R1013) was used for purification and samples were eluted in 12 μl H₂O. The structure of the final ligation product is indicated in Fig. S1A.

### Reverse transcription and library amplification

To generate cDNA, 200U Superscript II or III reverse transcriptase (Invitrogen, 18064014/18080044) was used with 40U of RNaseOUT (Invitrogen, 10777019) in a 20 μL volume without heat inactivation. The resulting cDNA was cleaned up using ExoSAP-IT (Applied Biosystems, 78201) enzymatic cleanup. After heat inactivation (15 min at 85 °C), the cDNA was used for real-time quantitative PCR (qPCR) with the Kapa HiFi HotStart Library Amp kit (Roche, KK2612). For Illumina short-read sequencing, libraries were amplified using dual indexed TruSeq i5/i7 primers until the fluorescence level reached standard 3 provided with the kit. For unoxidized samples with an initial RNA input of 1.5 μg per library, 11 cycles were commonly used. Amplified libraries were purified using the Zymoclean Gel DNA recovery kit (Zymo, D4008) from a 3% low range agarose gel (BioRad, 1613107) to remove adapter dimers.

## Computational/statistical analysis/data processing

Ready prepared libraries from RKO cells were sequenced using the Illumina HiSeqV4 SR50 mode. Libraries prepared from mouse samples were sequenced on an Illumina NovaSeq SP in SR100 XP mode and on an Illumina NovaSeq S4 in 150 paired-end mode (B cell and PC experiments). In the latter case, only read1 was further analyzed. Sequencing quality control was conducted with fastqc v 0.11.8. Small RNA sequencing reads obtained from RKO time-course samples were mapped to human genome GRCh38 and the microRNA and spike-ins counts were quantified with the annotation from miRbase (Kozomara et al, 2019), using the same NextFlow pipeline as in Dexheimer et al, 2020. Spike-in normalization was performed for miRNA expression levels with unit of amol/μg of total RNA.

For mouse samples, an updated version of this pipeline was used, in more detail: Raw reads were demultiplexed and analyzed with a NextFlow pipeline that orchestrated public bioinformatics tools and custom python scripts based on biopython, pysam, pandas, and numpy libraries (Di Tommaso et al, 2017). Raw reads were parsed and filtered as follows (Fig. S1B,C):

First, we aligned the expected adapter sequence ('AGATCG-GAAGAGCACACGTCT') to the read sequence using a global sequence alignment with matches scoring +1 and gap opening/extending penalties −2 and −1, respectively. Start/end gaps were not penalized. The cumulative alignment score was length-normalized, and reads were filtered ('no_adapter' reads) by comparing to a minimum score (0.9 in our experiments). Sequences upstream of the aligned adapter were extracted and interpreted as sRBC barcode (5 nt) and UMI (6 nt). Sequences upstream of the UMI were then extracted and interpreted as small RNA reads. Small RNA reads shorter than 18 nt ('too_short') and reads where the extracted sRBC barcode did not match the expectation ('wrong_srbc', e.g., due-to cross-sample contamination) were filtered. The resulting pre-filtered FASTQ files were then further processed by fastp (v0.20.1) which was configured to first trim 4nt off the read start and then filter reads based on quality (default settings) and a minimum remaining read length of 18nt (see Fig. S2 for example statistics).

We then filtered and counted reads stemming from our 8 different spike-in RNAs (see below) by aligning the spike-in sequences to the read, allowing for one mismatch. On average, we assigned ~2–3% of reads to the different spike-ins and filtered them from our datasets. Remaining small RNA reads were then mapped to a transcriptome created from MirGeneDB v2.0 annotations using Tailor v1.1 (Chou et al, 2015). Transcriptomes (FASTA and GFF3 annotation files) were created by extracting contigs from the human (GRCh38) and mouse (mm10) reference genomes using the respective pre-miRNA annotations, where overlapping annotations resulted in single contigs containing all respective annotations. Mappability tracks for visual inspection and QC were calculated with genmap v1.3.0 (Pockrandt et al, 2020) (parameters k = 18, e = 2). The minimum prefix matching parameter of Tailor (-l) was set to 18 nt. We then created down-sampled BAM files from the full alignments for visual inspection in a genome browser, debugging and QC purposes.

Finally, small RNA reads were counted by a custom python script. For pre-miRNA annotations, we counted strand-specific overlapping reads. For mature miRNAs, we counted only reads for which the 5′end is within +/−5 bp of the annotation 5′end and for which the 3′end does not exceed more that 5 bp beyond the annotation. Soft-clipped bases at the 3′ends of the reads were interpreted as tails, extracted, and counted. Counts of multi-mapping reads were normalized by the number of optimal alignment positions (1/NH). Resulting count tables for mouse samples were further analyzed in R using Rstudio v2022.12.0.

We estimated sequencing library sizes from concentration-normalized methylated spike-in read counts and then normalized

read counts by these values. To exclude miRNAs with low confidence of expression from our analysis, we implemented a threshold. For downstream analysis only miRNAs with reads >0.01 (human samples) or >0.5 (mouse samples) in spike-in normalized, unoxidized samples were considered. Consequently, this translates to fewer than 15 miRNA molecules in RKO cells and in mouse cells, considering the total RNA content per cell is approximately 10 picograms (pg) for RKO and about 1–4 pg for our mouse samples.

For results tables as used for generating graphs shown in this study, see Dataset EV1. Dataset EV2 contains PC miRNAs and miRNAs identified with high confidence in 1%, 0.1%, and 0.01% PC samples. Plots were generated using ggplot2 (v3.3.6) and GraphPad Prism (v10.0.2). Figures were prepared with Adobe Illustrator (v25.4.1). Further information about processing of small RNA libraries as described can be found in Appendix Figs. S1, 2.

## Data availability

Raw sequencing data are available as Zenodo repository (https://doi.org/10.5281/zenodo.10014351). Pipelines used for processing of small RNA sequencing data can be found on GitHub under https://github.com/lengfei5/smallRNA_nf/tree/master/dev_sRBC (used for RKO time-course) and under https://github.com/popitsch/pysrna (used for mouse B cell and plasma cell libraries, supports as well as samples from human cells). Raw flow cytometry data is available as BioStudies repository (https://doi.org/10.6019/S-BSST1360).

The source data of this paper are collected in the following database record: biostudies:S-SCDT-10_1038-S44318-024-00102-8.

## Peer review information

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

## Acknowledgements

We thank Raphael Manzenreither and Nina Khaldieh from the Ameres group for support with northern blot experiments and small RNA sequencing libraries. We thank Stephen F. Konieczny (Purdue University, West Lafayette, USA) for providing the Bhlha15$^{Cre/+}$ mouse line, C. Theussl's team for generating the R26$^{LSL-HenT6B/+}$ mouse line, K. Aumayr's team (IMP, Vienna) for flow-cytometric sorting, the VBCF NGS facility for all Illumina sequencing. This research was supported by Boehringer Ingelheim, and NSF CAREER Award 2238425 to L.C. Research in the Ameres group is supported by the European Research Council (ERC-CoG-866166) and the Austrian Science Fund FWF (SFB-F8002). For the purpose of open access, the author (S.A.) has applied a CC BY public copyright license to any Author Accepted Manuscript version arising from this submission.

## Author contributions

**Ariane Mandlbauer**: Conceptualization; Formal analysis; Investigation; Methodology; Writing—original draft; Writing—review and editing. **Qiong Sun**: Formal analysis; Investigation; Methodology. **Niko Popitsch**: Formal analysis. **Tanja Schwickert**: Formal analysis; Investigation; Methodology. **Miroslava Spanova**: Investigation. **Jingkui Wang**: Formal analysis. **Stefan L Ameres**: Conceptualization; Supervision; Funding acquisition; Writing—review and editing. **Meinrad Busslinger**: Conceptualization; Supervision; Funding acquisition; Writing—review and editing. **Luisa Cochella**: Conceptualization; Supervision; Funding acquisition; Writing—original draft; Writing—review and editing.

Source data underlying figure panels in this paper may have individual authorship assigned. Where available, figure panel/source data authorship is listed in the following database record: biostudies:S-SCDT-10_1038-S44318-024-00102-8.

## Disclosure and competing interests statement

SLA is co-founder, scientific advisor, and member of the board of QUANTRO Therapeutics GmbH. The other authors declare no competing interests.

# Expanded View Figures

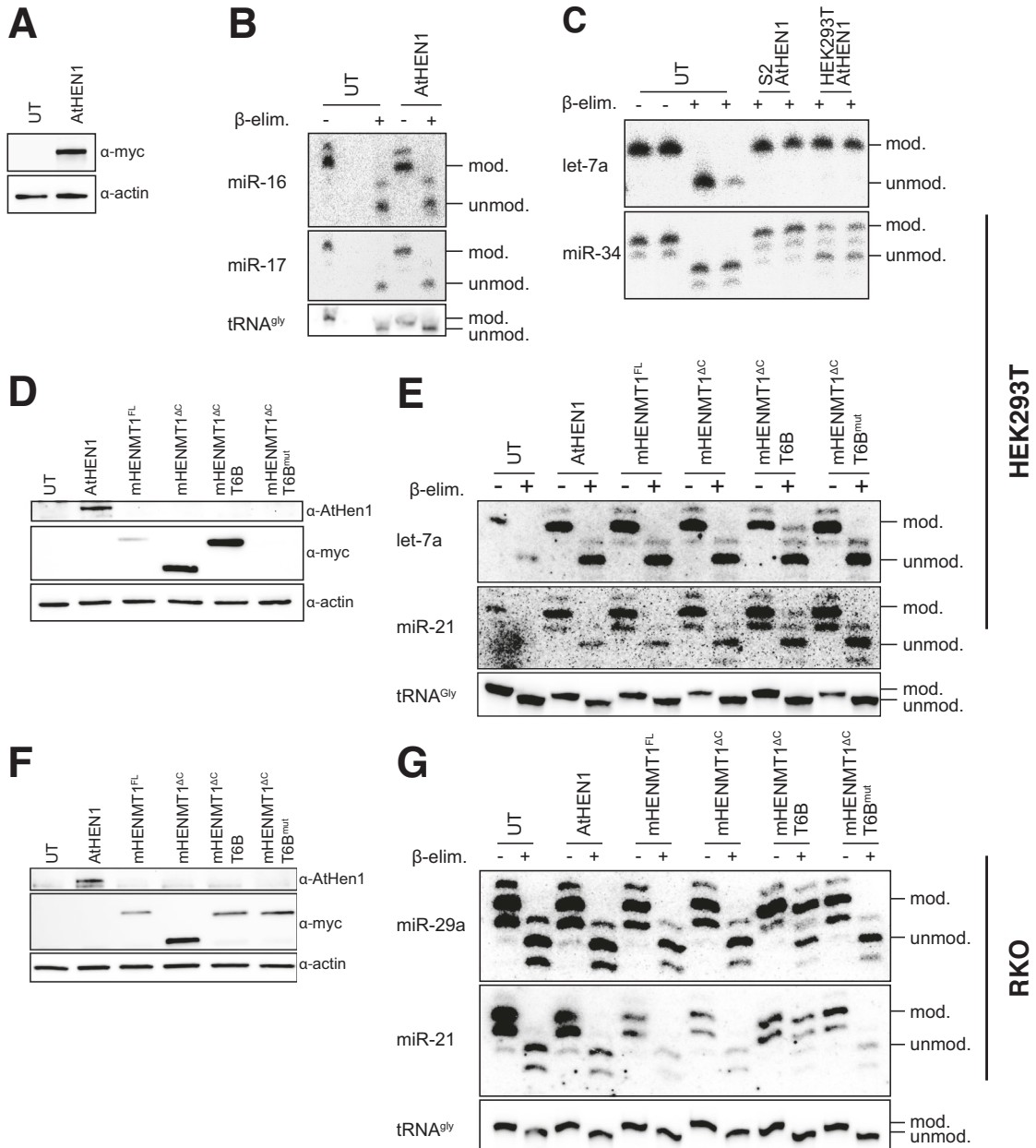

**Figure EV1. HENMT1^ΔC-T6B but not At-HEN1 efficiently methylates mouse and human miRNAs in cultured cells.**

(A–E) HEK293T cells transfected with a construct for expression of codon optimized and Myc-tagged At-HEN1. (A) Western blot for Myc-tagged At-HEN1 from HEK293T cells. Actin served as loading control. (B) Total RNA from the HEK293T cells was subjected to oxidation and β-elimination and the products resolved by high-resolution northern blotting with the indicated probes (tRNA^Gly was used to monitor oxidation completion, and as loading control). (C) In vitro methylation assay. FLAG-tagged At-HEN1 was immunopurified upon expression in S2 cells, or Myc-tagged At-HEN1 from HEK293T cells and incubated at 37 °C with radiolabeled dme-let-7 or dme-miR-34 duplex RNAs with 2-nt 3′ overhangs. Incubation without enzyme served as negative control. Methylation was assessed by β-elimination and high-resolution PAGE. (D) Western blots to monitor expression of the various enzymes in HEK293T cells transfected with indicated constructs. (E) Total RNA extracted from HEK293T cells transiently transfected with indicated constructs, and untransfected control (UT) was treated as in Fig. 1D. (F, G) RKO cells transduced with indicated constructs under a doxycycline inducible TRE3 in cells that have integrated the rtTA3 transactivator. (F) Western blots to monitor expression of the various enzymes with titrated doxycycline concentrations. (G) Northern blots as described in (B). Experiments were performed at least twice per cell line. Source data are available online for this figure.

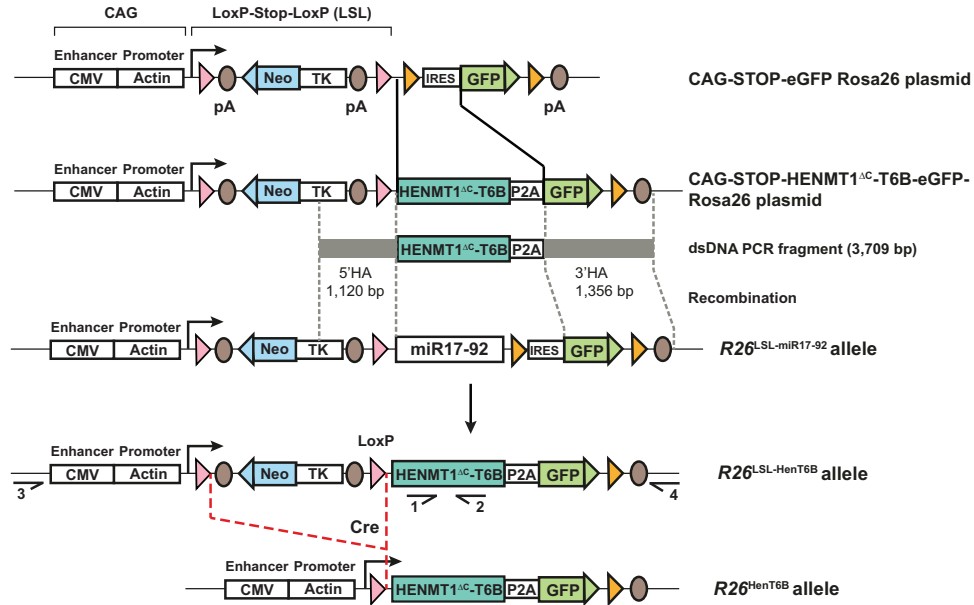

**Figure EV2.   Generation of the *R26*^LSL-HenT6B^ allele.**

The HENMT1^ΔC^-T6B cDNA was cloned in the CAG-STOP-eGFP-Rosa26 TV plasmid by replacing the IRES sequence with the HENMT1^ΔC^-T6B cDNA fused in frame via a P2A peptide to the eGFP coding sequence to generate the CAG-STOP-HENMT1^ΔC^-T6B-eGFP-Rosa26 plasmid. A 3,709-bp PCR fragment containing a 1120-bp 5′ homology arm (HA) and a 1356-bp 3′ HA was used as double-stranded DNA repair template together with two sgRNAs to replace the miR17-92-IRES DNA sequences of the *R26*^LSL-miR17-92^ allele[26] by CRISPR/Cas9-mediated genome editing in mouse 2-cell embryos, resulting in the *R26*^LSL-HenT6B^ allele. Cre-mediated deletion of the loxP-Stop-loxP (LSL) cassette leads to the expression of the HENMT1^ΔC^-T6B-P2A-eGFP gene from the ubiquitously transcribed CAG promoter of the *R26*^HenT6B^ allele. *LoxP* and *frt* sites are indicated by red and yellow arrowheads, respectively. The herpes simplex virus thymidine kinase (TK) promoter drives expression of the neomycin (Neo) resistance gene. Arrows indicate primers 1 and 2 used for genotyping of the *R26*^LSL-HenT6B^ allele and primers 3 and 4 used for genotyping of the wild-type *R26* allele (see Methods). pA, polyadenylation sequence.

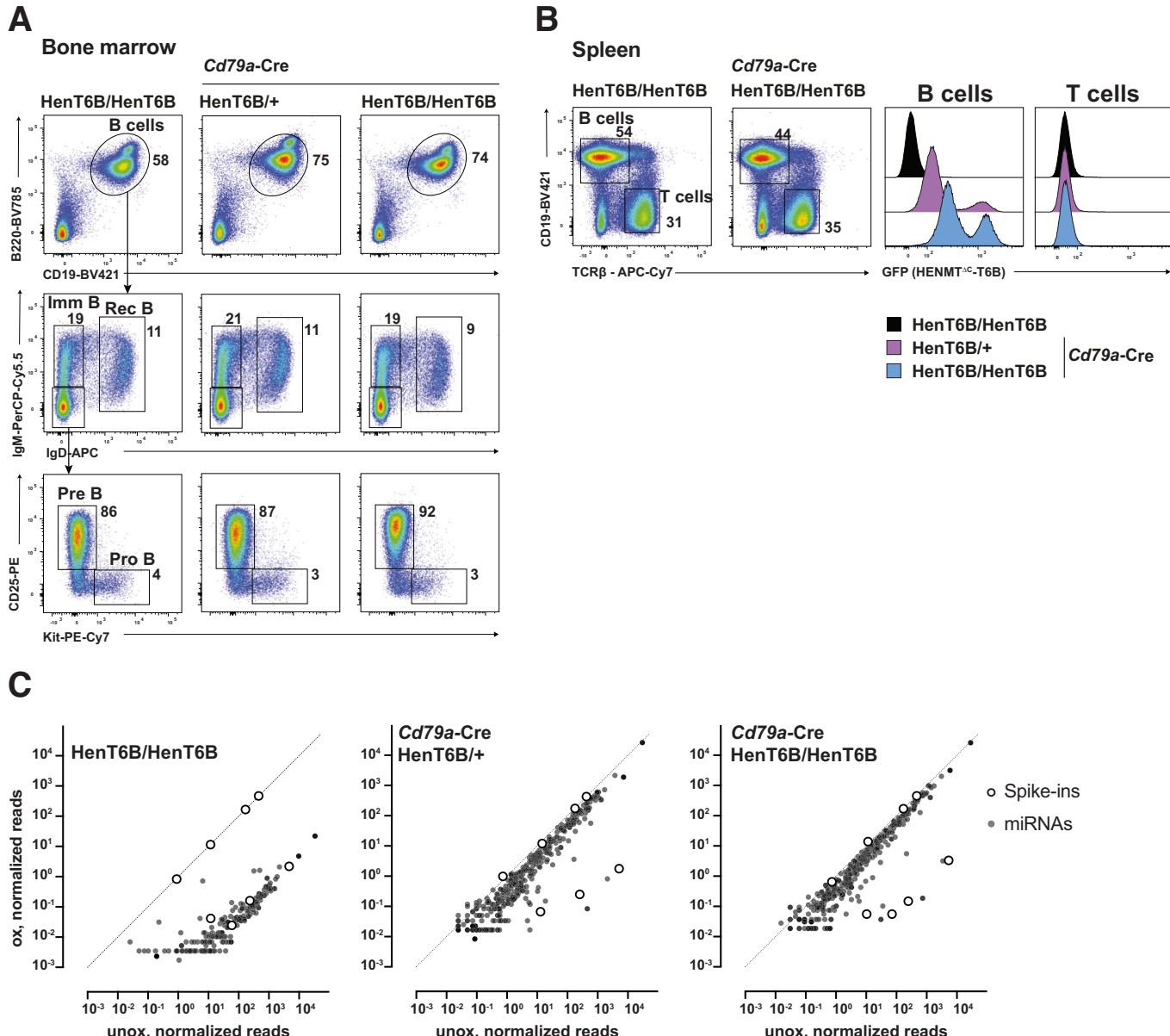

**Figure EV3.   Analysis of B cell development in *Cd79a*-Cre *R26*[LSL-HenT6B/LSL-HenT6B] mice.**

(**A**) Flow-cytometric analyses of the indicated B cell types in the bone marrow of *Cd79a*-Cre *R26*[LSL-HenT6B/+], *Cd79a*-Cre *R26*[LSL-HenT6B/LSL-HenT6B] and control *R26*[LSL-HenT6B/LSL-HenT6B] mice at the age of 6–8 weeks. The percentage of cells in the indicated gates is shown. One of three independent experiments is shown. (**B**) Flow-cytometric analysis of B and T cells in the spleen of *Cd79a*-Cre *R26*[LSL-HenT6B/+], *Cd79a*-Cre *R26*[LSL-HenT6B/LSL-HenT6B] and control *R26*[LSL-HenT6B/+] or *R26*[LSL-HenT6B/LSL-HenT6B] mice at the age of 6–8 weeks. The percentage of B and T cells is shown next to the respective gate (left). The GFP expression of B and T cell of the indicated genotypes is shown as a histogram (right). One of three independent experiments is shown. (**C**) Total RNA was extracted from B cells of the indicated genotypes (see Methods for purification strategy), mixed with four methylated and four unmethylated RNA spike-ins and subjected to mime-seq. Libraries were normalized to methylated spike-ins. Normalized reads of oxidized vs. unoxidized libraries are plotted for every miRNA with > 0.5 norm. reads. Note the depletion of unmethylated miRNAs (similar to the four unmethylated spike-ins) and the strong enrichment when HENMT1[ΔC]-T6B is expressed (close to the four methylated spike-ins).

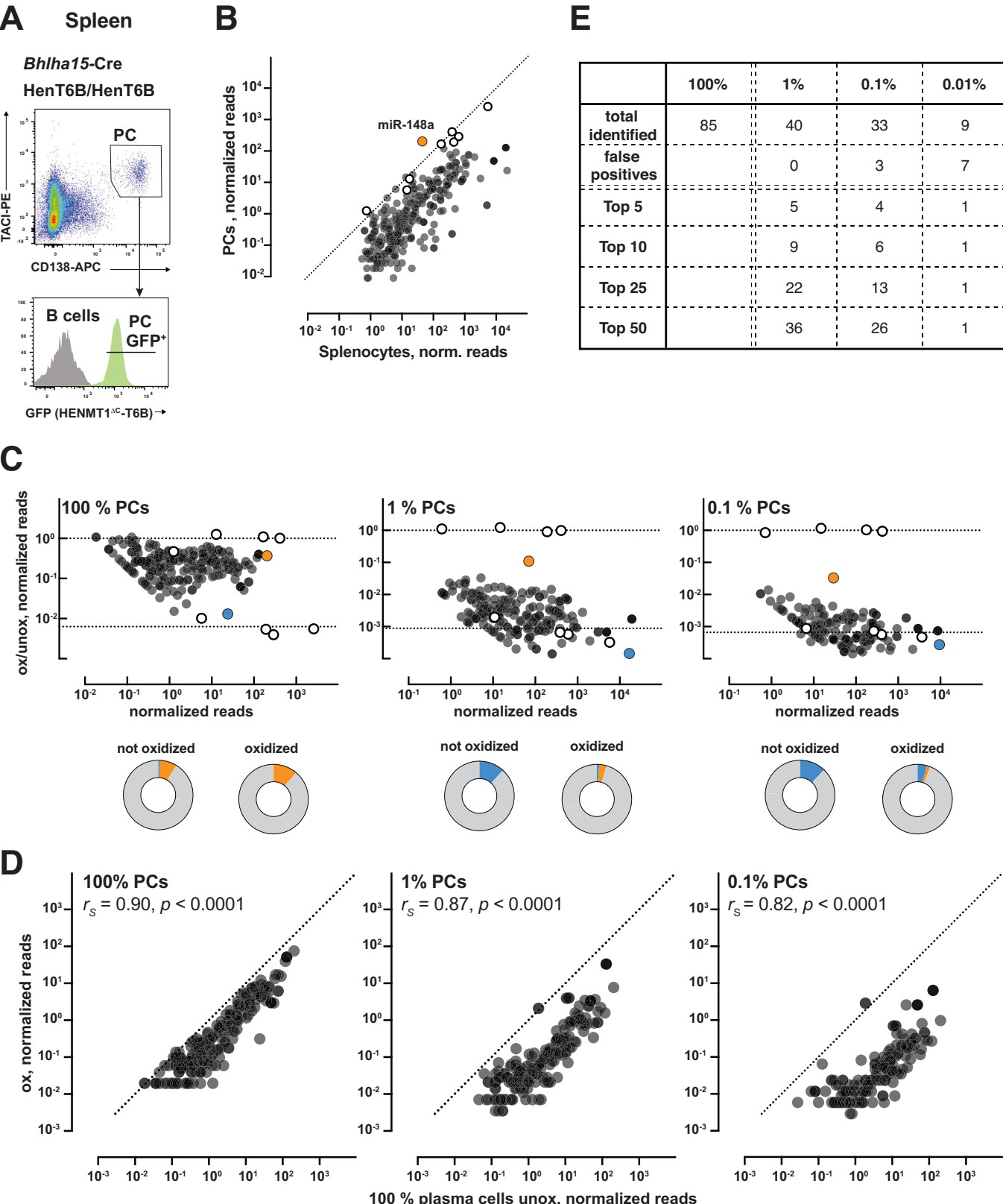

◀ **Figure EV4.    Mime-seq identifies cell-specific miRNAs with high sensitivity and maintains relative abundance of miRNAs.**

(A) Flow-cytometric sorting of plasma cells from the spleen of *Bhlha15*-Cre *R26*[LSL-HenT6B/LSL-HenT6B] mice at day 7 after immunization with sheep red blood cells. The sorting gates used for the isolation of plasma cells (CD138[+]TACI[+]) are indicated. (B) Comparison of unoxidized, spike-in normalized small RNA libraries from 100% PCs vs 99.99% splenocytes (0.01% PCs) showed that miR-148a is exclusively expressed in PCs. (C) Fractions recovered for all sequenced miRNAs with normalized reads >0.5 (PC mixing ratios indicated above). Pie charts below show proportions of miRNA reads from oxidized and unoxidized samples, highlighting the enrichment of miR-148 (orange) upon oxidation even from samples with 0.1% PCs, as well as the depletion of miR-451 (blue) from contaminating erythrocytes. (D) Oxidized libraries from samples with the indicated mixing ratios vs. normalized reads from unoxidized pure PCs. The relative abundance of miRNAs present in PCs is maintained after oxidation, showing that mime-seq remains semi-quantitative down to 0.1% PC ($r_S$ = Spearman correlation coefficient). (E) Summary table of the most highly abundant PC miRNAs recovered in 1%, 0.1%, or 0.01% PCs.

