## [Peer Review File · The EMBO Journal]

Mime-seq 2.0: a method to sequence microRNAs from specific mouse cell types

Ariane Mandlbauer, Qiong Sun, Niko Popitsch, Tanja Schwickert, Miroslava Spanova, Jingkui Wang, Stefan Ameres, Meinrad Busslinger, and Luisa Cochella

Corresponding authors: Luisa Cochella (mcochel1@jhmi.edu) , Stefan Ameres (stefan.ameres@univie.ac.at), Meinrad Busslinger (Meinrad.Busslinger@imp.ac.at)

Review Timeline:

Submission Date:	4th Dec 23
Editorial Decision:	6th Feb 24
Revision Received:	12th Mar 24
Editorial Decision:	27th Mar 24
Revision Received:	29th Mar 24
Accepted:	2nd Apr 24

Editor: Yehu Moran

Transaction Report:

Dear Dr. Cochella,

Thank you for submitting your manuscript for consideration by the EMBO Journal. It has now been seen by three expert referees whose comments are provided below.

Given the referees' positive recommendations, I would like to invite you to submit a revised version of the manuscript, addressing in detail the comments of all three reviewers. I should add that it is EMBO Journal policy to allow only a single round of revision, and acceptance of your manuscript will therefore depend on the completeness of your responses in this revised version.

Thank you for the opportunity to consider your work for publication. I look forward to your revision.

Yours sincerely,

Yehu Moran
Academic Editor
The EMBO Journal

Please make sure you upload a letter of response to the referees' comments together with the revised manuscript. Please make sure to address all comments in detail.

Please remember: Digital image enhancement is acceptable practice, as long as it accurately represents the original data and conforms to community standards. If a figure has been subjected to significant electronic manipulation, this must be noted in the figure legend or in the 'Materials and Methods' section. The editors reserve the right to request original versions of figures and

the original images that were used to assemble the figure.

We realize that it is difficult to revise to a specific deadline. In the interest of protecting the conceptual advance provided by the work, we recommend a revision within 3 months (6th May 2024). Please discuss the revision progress ahead of this time with the editor if you require more time to complete the revisions.

Referee #1:

In this manuscript, A. Mandlbauer and colleagues describe an improvement of the "mime-seq" technique. That technique allows cell-specific sequencing of miRNAs without the need for cell sorting (rather using an elegant method based on miRNA methylation, where the methylase is expressed in the cell type of interest; sequencing methods allow the selective detection of methylated miRNAs only, therefore enriching for miRNAs from that cell type). This is a much-needed method, addressing most of the issues the community is facing with single-cell sequencing in Small RNA-Seq - and therefore the topic is of high technical interest. The original technique (published by the same labs in 2018) was developed for *D. melanogaster* and *C. elegans*, but it did not work in mammalian cells. The current manuscript describes an adaptation of the method for mammalian cells (basically, a mammalian methylase has to be tethered to the Argonaute partner of miRNAs for the methylation reaction to work efficiently in mammalian cells). The manuscript is well written, it is very clear and the data is convincing. I could only spot a few minor defects, which should be easy to fix before the manuscript is acceptable for publication.

Minor points:

1. Figure 1D: It is not fully clear what the "+" and "-" signs indicate (with or without oxidation treatment?). Please clarify.
2. Figure 2C: axis legends are very confusing. I cannot make sense of the "-" signs, they don't appear to be "minus" signs. What do the authors mean when they write "normalized reads - unox" and "UT - log₁₀(normalized reads)"? If this sign is just a meaningless delimiter (meaning something like "normalized reads, unox") then the axis titles would make much more sense; if it is the case, then please use another symbol than a minus sign.
3. Legend for Figures 2 B-E: "MicroRNAs with spike-in-normalized reads <0.01 were removed from the analysis" (same thing on main text, line #160). This is quite obscure. What does this number represent exactly? The amount of introduced spike-in oligos should be given (I could not find it), and if this "0.01" value is a cutoff value on the ratio between read numbers for miRNAs and one of the spike-ins (or the sum of all of them?), then please say it explicitly.
4. Figure 2E: the figure legend mentions a correlation coefficient, which does not appear on the plots.
5. Figures 3 B and C: "Statistical data (B,C) were analyzed by one-way ANOVA with Dunnett's multiple comparison test". The method employed here makes perfect sense, but one piece of information is missing: which condition has been used as a control in Dunnett's test? Intuitively I would have expected it to be the HenT6B/HenT6B condition (black bars), but this is not consistent with the fact that some of the asterisks shown span only the purple and blue bars. Please clarify.
6. Materials and Methods, section "RNA extraction and preparation": "sequences are provided in supplementary table X", with X being an apparent placeholder that needs to be corrected.

Referee #2:

Mandlbauer et al.

Mime-seq 2.0: a method to sequence microRNAs from specific mouse cell types

In this manuscript, the authors have further developed a tissue-specific miRNA sequencing method called Mime-seq. This is an elegant method that uses the plant methyltransferase Hen1 in *C. elegans* and flies to methylate miRNAs. This, however, did not work for mammalian cells. Therefore, the authors used HENMT1 that is naturally expressed in mouse germ cells and functions in the piRNA pathway. They generated a C-terminally truncated version and fused it to an Argonaute binding peptide termed T6B. Using this approach, the methyltransferase is targeted to Ago proteins and methylates the bound miRNA. An adapted sequencing protocol that oxidizes non-methylated miRNAs can be used to specifically detect the modified miRNAs. This approach worked well in tissue culture cells and thus the authors produced an inducible mouse allowing for the expression of the

T6B-HENMT1 in specific tissues and particularly in rare tissues and cells. The authors investigated a number of rare cells from the hematopoietic lineage and were able to enrich and identify cell-type specific miRNAs with high specificity. Even highly diluted cell numbers led to an accurate retrieval of the cell-specific miRNAs.

This is a very clear and straightforward study. It is well presented and provides an elegant tool for the field. Particularly the mouse model will be highly appreciated and allows for accurate miRNA profiling without prior cell sorting. I did not find any technical issues or problems with this study. I have only one point/comment that should be considered.

The fact that overexpression of the T6B-HENMT1 in the B cell lineage does not cause any effects on B cell development is somewhat surprising. The T6B peptide should lead to a global miRNA inhibition and this has been associated with hematopoietic cell development. However, an explanation could be that the T6B-fused enzyme is only inefficiently bound to Ago proteins but nevertheless methylates miRNAs even in an Ago-unbound state. To control for that, a non-fused HENMT1 should be overexpressed. A mutated and Ago-binding-deficient T6B peptide could also be used. This should at least be discussed in detail.

Referee #3:

In the manuscript, the authors describe mime-seq2, a variant of their previously described approach based on artificially methylating miRNAs within cells by expressing a particular transgene and then reading out the methylation status of miRNAs post-extraction. Overall, this is a very clever and efficient approach. Mime-seq was developed for nematode and fly cells, and here the authors mention that the original approach does not work in mammalian cells, likely due to the more limited availability of "naked" Dicer products. Instead, they utilize an alternative methylation protein that they fuse to an Ago-interacting peptide and show in cells and in a mouse model that this approach can indeed methylate miRNAs in culture and in blood cells and can be indeed used to identify cell-type-specific miRNAs expressed in a small subpopulation of cells in vivo, given that a suitable Cre driver is available for that population.

The paper is very well written, and the results are carefully and informatively presented. The approach is clever, though the innovation is primarily technical rather than conceptual, as the concept was already introduced in mime-seq. My only yet major concern is that I strongly wonder if the method will be used by more than a handful of other labs. The authors correctly mention that the breeding scheme for introducing the methylating peptide is relatively simple. Still, any breeding of mouse models introduces considerable work and costs, and if breeding is needed, potential users can just breed their mice to a Cre-driven GFP model, and sort and sequencing the positive cells. This is bound to give more information as spatial examination of the GFP signal will also be available, other RNA types can be sequenced, etc. As the authors note, their method is only semi-quantitative, and so mostly relevant in cases where a particular miRNA is really cell-type-specific, but in these cases it will be easy to study also by sequencing sorted cells. Also, most functionally relevant miRNAs in mice are already probably known, and I doubt that mime-seq2 will be able to find many additional and highly specific miRNAs. The results presented in the manuscript also do not really show a use-case scenario where mime-seq2 is giving insights that were not previously accessible. As such, this manuscript is mostly a technical tour-de-force, and sets a high technical standard, but its impact on the miRNA community will likely be limited, and as such it might be more suitable for a more specialized journal than EMBO J.

Point by point response to the reviewers: our replies are in blue

We thank all three reviewers for their time, comments and advice on how to improve our work.

Referee #1:

In this manuscript, A. Mandlbauer and colleagues describe an improvement of the "mime-seq" technique. That technique allows cell-specific sequencing of miRNAs without the need for cell sorting (rather using an elegant method based on miRNA methylation, where the methylase is expressed in the cell type of interest; sequencing methods allow the selective detection of methylated miRNAs only, therefore enriching for miRNAs from that cell type). This is a much-needed method, addressing most of the issues the community is facing with single-cell sequencing in Small RNA-Seq - and therefore the topic is of high technical interest. The original technique (published by the same labs in 2018) was developed for *D. melanogaster* and *C. elegans*, but it did not work in mammalian cells. The current manuscript describes an adaptation of the method for mammalian cells (basically, a mammalian methylase has to be tethered to the Argonaute partner of miRNAs for the methylation reaction to work efficiently in mammalian cells). The manuscript is well written, it is very clear and the data is convincing. I could only spot a few minor defects, which should be easy to fix before the manuscript is acceptable for publication.

Minor points:

1. Figure 1D: It is not fully clear what the "+" and "-" signs indicate (with or without oxidation treatment?). Please clarify.

We apologize for the lack of a clear annotation in the figure. The signs referred to whether the sample had been oxidized and beta-eliminated ("+") or not ("-"). This information has now been added to the revised figure 1.

2. Figure 2C: axis legends are very confusing. I cannot make sense of the "-" signs, they don't appear to be "minus" signs. What do the authors mean when they write "normalized reads - unox" and "UT - log₁₀(normalized reads)"? If this sign is just a meaningless delimiter (meaning something like "normalized reads, unox") then the axis titles would make much more sense; if it is the case, then please use another symbol than a minus sign.

We see the problem with our previous labeling and the dash. The axes have been relabeled to indicate the relevant comparison in that figure: non-transduced vs. HENMT1-T6B-expressing cells, all under unoxidized conditions (stated in the legend).

3. Legend for Figures 2 B-E: "MicroRNAs with spike-in-normalized reads <0.01 were removed from the analysis" (same thing on main text, line #160). This is quite obscure. What does this number represent exactly? The amount of introduced spike-in oligos should be given (I could not find it), and if this "0.01" value is a cutoff value on the ratio between read numbers for miRNAs and one of the spike-ins (or the sum of all of them?), then please say it explicitly.

We appreciate the reviewer highlighting this ambiguity. To exclude miRNAs with low confidence of expression (and thus considered biologically insignificant) from our analysis, we implemented a threshold. Specifically, we set a cutoff at 0.01 normalized counts based on spike-in normalized reads in RKO cells and 0.5 in all mouse samples. This cutoff, informed by our spike-in normalization (detailed in the note on spike-in concentrations), equates to 2 attomoles (amol) of miRNA per microgram (μg) of total RNA. Consequently, this translates to fewer than 15 miRNA molecules in RKO cells and in our mouse samples, considering the total RNA content per cell is approximately 10 picograms (pg) for RKO and about 1-4 pg for our mouse samples. These figures are in stark contrast to the most abundant miRNAs, such as miR-21-5p, which exists in approximately 42,000 copies per RKO cell, demonstrating that our chosen threshold is conservative.

We have added this rationale to the methods section (lines 506-510) and have updated Table EV1 with the spike-in concentrations utilized and have specified the spike-in

oligonucleotides employed in the Methods section. The range of spike-in concentrations spans from the lowest at 10 amol/ μg of total RNA (approximately 60 molecules per cell) to the highest at 5,000 amol/ μg of total RNA (approximately 30,110 molecules per cell).

4. Figure 2E: the figure legend mentions a correlation coefficient, which does not appear on the plots.

We apologize for the oversight. The correlation coefficients have been reinstated in the figure.

5. Figures 3 B and C: "Statistical data (B,C) were analyzed by one-way ANOVA with Dunnett's multiple comparison test". The method employed here makes perfect sense, but one piece of information is missing: which condition has been used as a control in Dunnett's test? Intuitively I would have expected it to be the HenT6B/HenT6B condition (black bars), but this is not consistent with the fact that some of the asterisks shown span only the purple and blue bars. Please clarify.

We thank the reviewer for pointing out this inconsistency. By mistake, we compared all data with the B cell type of the heterozygous (*Cd79a*-Cre HenT6B/+) genotype. In the meantime, we have performed two additional flow-cytometric experiments with heterozygous (*Cd79a*-Cre HenT6B/+), homozygous (*Cd79a*-Cre HenT6B/HenT6B) and control (HenT6B/+ or HenT6B/HenT6B) mice. We now examined at least 10 mice of each genotype in 3 independent experiments, which has significantly improved the statistical analysis. We have analyzed the new flow-cytometric data by comparing the data relative to the control genotypes (HenT6B/+ or HenT6B/HenT6B) with one-way ANOVA with Dunnett's multiple comparisons test. Due to the analysis of more mice, each B cell type is now shown to be present at similar frequency in the bone marrow of the different genotypes.

6. Materials and Methods, section "RNA extraction and preparation": "sequences are provided in supplementary table X", with X being an apparent placeholder that needs to be corrected.

Indeed, the reference to "Table X" was intended to direct readers to Table EV1. This has now been corrected.

Referee #2:

Mandlbauer et al.

Mime-seq 2.0: a method to sequence microRNAs from specific mouse cell types

In this manuscript, the authors have further developed a tissue-specific miRNA sequencing method called Mime-seq. This is an elegant method that uses the plant methyltransferase Hen1 in *C. elegans* and flies to methylate miRNAs. This, however, did not work for mammalian cells. Therefore, the authors used HENMT1 that is naturally expressed in mouse germ cells and functions in the piRNA pathway. They generated a C-terminally truncated version and fused it to an Argonaute binding peptide termed T6B. Using this approach, the methyltransferase is targeted to Ago proteins and methylates the bound miRNA. An adapted sequencing protocol that oxidizes non-methylated miRNAs can be used to specifically detect the modified miRNAs. This approach worked well in tissue culture cells and thus the authors produced an inducible mouse allowing for the expression of the T6B-HENMT1 in specific tissues and particularly in rare tissues and cells. The authors investigated a number of rare cells from the hematopoietic lineage and were able to enrich and identify cell-type specific miRNAs with high specificity. Even highly diluted cell numbers led to an accurate retrieval of the cell-specific miRNAs.

This is a very clear and straightforward study. It is well presented and provides an elegant tool for the field. Particularly the mouse model will be highly appreciated and allows for accurate miRNA profiling without prior cell sorting. I did not find any technical issues or problems with this study. I have only one point/comment that should be considered.

The fact that overexpression of the T6B-HENMT1 in the B cell lineage does not cause any effects on B cell development is somewhat surprising. The T6B peptide should lead to a global miRNA inhibition and this has been associated with hematopoietic cell development. However, an explanation could be that the T6B-fused enzyme is only inefficiently bound to Ago proteins but nevertheless methylates miRNAs even in an Ago-unbound state. To control for that, a non-fused HENMT1 should be overexpressed. A mutated and Ago-binding-deficient T6B peptide could also be used. This should at least be discussed in detail.

The reviewer highlights an important point regarding a seeming discrepancy between a previously published study, in which overexpression of a T6B-YFP fusion protein caused an increase in pro-B cell numbers and a strong decrease in pre-B, immature and mature B cell numbers in the bone marrow of transgenic mice (La Rocca *et al.* *eLife*, 2021). In marked contrast, we observed similar numbers of each B cell type in the bone marrow of control (HenT6B/+ or HenT6B/HenT6B), heterozygous (*Cd79a-Cre* HenT6B/+) and homozygous (*Cd79a-Cre* HenT6B/ HenT6B) mice (revised Figure 3B,C). La Rocca *et al.* proposed that T6B as part of T6B-YFP fusion protein acts as a competitive inhibitor of TNRC6-dependent gene regulation mediated by Ago-bound miRNAs. As such, the efficacy, with which T6B can compete with endogenous Ago-TNRC6 interactions, may be contingent on the extent of T6B overexpression and/or its affinity for Ago. It is plausible that the lower expression of HenT6B from the *Rosa26* locus (our study) compared to the doxycycline-induced expression of T6B-YFP from the *Col1a1* locus (La Rocca *et al.*, 2021) might explain the discrepancy of the observed B cell phenotypes. We have now included this consideration in a new paragraph within the discussion section (lines 261-271).

In response to the reviewer's suggestion to test miRNA methylation activity of non-T6B-fused HENMT1 variants, we examined the modification status of endogenous miRNAs upon expression of full-length HENMT1 and the truncated variant (HENMT1^{ΔC}) – neither of which were fused to T6B – across three different cell types. In neither condition did we observe miRNA methylation (as shown in Figures 1 and EV1), strongly suggesting that miRNA methylation requires T6B-dependent association of HENMT1 with Ago. These findings and their implications are now detailed in lines 113-117 and 248-250 in our manuscript.

New paragraph for discussion:

When testing mime-seq 2.0 in B cell types of the mouse hematopoietic system, we found no indication of inhibition of miRNA function by HENMT1^{ΔC}-T6B. Overexpression of a T6B-YFP fusion

protein was, however, previously shown to inhibit miRNA function by competing for binding of endogenous TNRC6 to Argonaute, although this did not affect the miRNA repertoire (La Rocca *et al.*, 2021). Within the hematopoietic system, T6B-YFP overexpression under the control of a doxycycline-inducible promoter in the *Col1a1* locus resulted in an increase of pro-B cells and a decrease of pre-B, immature and mature B cells in the bone marrow (La Rocca *et al.*, 2021). In contrast, we observed similar frequencies of all B cell types in the bone marrow upon B cell-specific expression of HENMT1^{ΔC}-T6B from the *Rosa26* locus. Given that T6B acts as a competitive inhibitor, this discrepancy might be explained by differences in expression level, or binding affinity. In our study, the expression of HENMT1^{ΔC}-T6B seems to be sufficient to induce the necessary levels of miRNA methylation without affecting the generation or viability of the cells expressing the enzyme in contrast to the overexpression of T6B-YFP protein.

Referee #3:

In the manuscript, the authors describe mime-seq2, a variant of their previously described approach based on artificially methylating miRNAs within cells by expressing a particular transgene and then reading out the methylation status of miRNAs post-extraction. Overall, this is a very clever and efficient approach. Mime-seq was developed for nematode and fly cells, and here the authors mention that the original approach does not work in mammalian cells, likely due to the more limited availability of "naked" Dicer products. Instead, they utilize an alternative methylation protein that they fuse to an Ago-interacting peptide and show in cells and in a mouse model that this approach can indeed methylate miRNAs in culture and in blood cells and can be indeed used to identify cell-type-specific miRNAs expressed in a small subpopulation of cells in vivo, given that a suitable Cre driver is available for that population.

The paper is very well written, and the results are carefully and informatively presented. The approach is clever, though the innovation is primarily technical rather than conceptual, as the concept was already introduced in mime-seq. My only yet major concern is that I strongly wonder if the method will be used by more than a handful of other labs. The authors correctly mention that the breeding scheme for introducing the methylating peptide is relatively simple. Still, any breeding of mouse models introduces considerable work and costs, and if breeding is needed, potential users can just breed their mice to a Cre-driven GFP model, and sort and sequencing the positive cells. This is bound to give more information as spatial examination of the GFP signal will also be available, other RNA types can be sequenced, etc. As the authors note, their method is only semi-quantitative, and so mostly relevant in cases where a particular miRNA is really cell-type-specific, but in these cases it will be easy to study also by sequencing sorted cells. Also, most functionally relevant miRNAs in mice are already probably known, and I doubt that mime-seq2 will be able to find many additional and highly specific miRNAs. The results presented in the manuscript also do not really show a use-case scenario where mime-seq2 is giving insights that were not previously accessible. As such, this manuscript is mostly a technical tour-de-force, and sets a high technical standard, but its impact on the miRNA community will likely be limited, and as such it might be more suitable for a more specialized journal than EMBO J.

We appreciate the reviewer's opinion but respectfully disagree with some of the assessments based on the findings of our study:

Regarding the point that potential users may prefer to cross their specific Cre line to an inducible GFP transgene (such as Rosa26(LSL-GFP/+)), we would like to point out that our HENMT-expressing transgene also expresses GFP in a Cre-dependent manner. Hence, in addition to any advantage of visualizing transgene expression and facilitating cell sorting for miRNA sequencing, the HenT6B transgene further allow for miRNA analysis by mime-seq. We realize that the fact that our transgene includes a GFP reporter was not sufficiently pointed out. Therefore, we now mention this more explicitly in the text (lines 157-159). It is true that scientists working in the miRNA field, who have no access to mouse genetics expertise, will neither use our Rosa26(LSL-HenT6B-P2A-GFP/+) or the Rosa26(LSL-GFP/+) mouse strains. But for scientists with access to a mouse facility, the HenT6B transgene described here provides a considerable advantage.

The main advantage of our approach is that sorting will not be needed for miRNA sequencing. This is of significant advantage for a number of reasons: First, the enzymatic and mechanical treatments necessary for dissociation can introduce unknown changes due to cell stress, especially for tissue-embedded cells which may be damaged by this procedure. Second, for rare cell types, acquiring the necessary number of cells for high-quality libraries may require lengthy or even multiple sorting sessions. Finally, sorting requires access to a flow cytometer capable of sorting. Mime-seq overcomes these challenges and may provide a preferred solution relative to isolating GFP-positive cells.

We agree with the reviewer that most miRNAs in the mouse genome are most likely comprehensively annotated. We therefore do not expect that mime-seq will necessarily lead to the discovery of new miRNAs. However, even in the worm, where all miRNAs are definitely known, elucidating the precise patterns of expression and the specificity within a given tissue has been essential to fully understand their function. In general, miRNA profiling in the mouse is still performed at a surprisingly low resolution (complex tissues or even whole organs). We therefore anticipate that having a tool to gain such resolution will be useful for dissecting the function of even well-known miRNAs in different cell types.

Finally, only time will tell how broadly this mouse model will be implemented in the future. However, given comments from the other reviewers and the fact that we already received inquiries to ship the Rosa26(LSL-HenT6B-2A-GFP/+) mouse strain, we anticipate that this will be a useful new tool for the miRNA field.

Dear Dr. Cochella,

I am pleased to inform you that after going over your revised manuscript and your point-by-point response to the referees, I find your manuscript suitable for acceptance in principle. However, there are still few relatively small things that were caught by our editorial assistance team and need your attention (please see the list below my signature). Please make these corrections and re-submit your manuscript so we can formally accept it.

Congratulations and best wishes,
Yehu Moran

Academic Editor
EMBO Journal

Notes from editorial assistant:

- *AUTHORS: three corresponding authors => please ask them to confirm
- *DATA AVAILABILITY SECTION: in, should be moved to the end of Materials and Methods
- *FUNDING: Should Boehringer Ingelheim be listed among the funders in our system?
- *DATASET EV LEGENDS: 2 EV datasets are in, both need legends added to the files in a separate sheet
- *SOURCE DATA: in with completed checklist, Fig 3B and C fcs files were deposited; see sticky note; seems complete. Source data files should be uploaded as one (zipped) file per figure.
- *SYNOPSIS IMAGE: in but we need it as a jpg or png file and sized 550 pixels wide x 200 - 600 pixels high
- *SYNOPSIS TEXT: not provided, please provide

Additional Notes:

- Table EV1 is quite large and should also be made an EV Dataset.
- Figure EV2 only has one panel, so the label "A" should be removed from the figure file and the legend and from the callouts
- Categories are missing

- Figure legends:
 1. Please note that information related to n is missing in the legends of figures 3e; 4b. Please correct.
 2. Please note that the error bars are not defined in the legends of figures 3e; 4b. Please correct.

General instructions for preparing your revised manuscript (many of these were already addressed in your current version):

We realize that it is difficult to revise to a specific deadline. In the interest of protecting the conceptual advance provided by the work, we recommend a revision within 3 months (25th Jun 2024). Please discuss the revision progress ahead of this time with the editor if you require more time to complete the revisions.

The authors addressed the minor editorial issues.

Dear Dr. Cochella,

I am pleased to inform you that your manuscript has been accepted for publication in the EMBO Journal.

Yours sincerely,

Yehu Moran
Academic Editor
The EMBO Journal
